

# Entanglement negativity in flat holography

**Debarshi Basu, Ashish Chandra, Himanshu Parihar and Gautam Sengupta⋆**

Department of Physics, Indian Institute of Technology, Kanpur, 208 016, India

⋆ sengupta@iitk.ac.in

## Abstract

We advance holographic constructions for the entanglement negativity of bipartite states in a class of $(1+1)$−dimensional Galilean conformal field theories dual to asymptotically flat three dimensional bulk geometries described by Einstein Gravity and Topologically Massive Gravity. The construction involves specific algebraic sums of the lengths of bulk extremal curves homologous to certain combinations of the intervals appropriate to such bipartite states. Our analysis exactly reproduces the corresponding replica technique results in the large central charge limit. We substantiate our construction through a semi classical analysis involving the geometric monodromy technique for the case of two disjoint intervals in such holographic Galilean conformal field theories



# 1 Introduction

In recent years quantum entanglement has emerged as a fundamental issue connecting diverse areas of physics from many-body condensed matter systems to black holes and quantum gravity. It is well known in quantum information theory that bipartite pure state entanglement is characterized by the entanglement entropy which is the von Neumann entropy of the corresponding reduced density matrix. However the entanglement entropy is not a valid measure for mixed state entanglement due to contributions from irrelevant correlations. To address this significant issue several entanglement and correlation measures were introduced in quantum information theory. However most of these were not easily computable as they

involved extremization over LOCC protocols. Vidal and Werner [1] in a classic work introduced a computable measure for mixed state entanglement termed *entanglement negativity* (logarithmic negativity) which was defined as the trace norm of the partial transpose of the density matrix with respect to one of the subsystems and provided an upper bound to the distillable entanglement. Despite its non-convexity [2], entanglement negativity was proved to be an entanglement monotone and is widely used to characterize mixed state entanglement.

For extended quantum many-body systems with infinite degrees of freedom such entanglement measures are usually computationally intractable although a formal definition may be attempted. Significantly, it was shown in [3,4] that the entanglement entropy of bipartite states in $(1+1)$-dimensional relativistic conformal field theories ($\text{CFT}_{1+1}$) may be explicitly computed through a replica technique. Remarkably the replica technique described above could also be modified to compute the entanglement negativity of bipartite states in such relativistic $\text{CFT}_{1+1}$ described in [5–7].

Over the last decade there has been intense focus on the holographic characterization of entanglement in conformal field theories dual to bulk AdS geometries in the framework of the AdS/CFT correspondence [8]. This was pioneered by the classic work of Ryu and Takayanagi (RT) in [9,10] where it was conjectured that the universal part of the entanglement entropy of a subsystem in a relativistic $\text{CFT}_d$ was proportional to the area of a bulk static codimension two minimal surface homologous to the subsystem. A covariant generalization of the above holographic conjecture was proposed by Hubeny, Rangamani and Takayangi (HRT) for relativistic $\text{CFT}_d$ dual to bulk non-static AdS geometries in [11]. The above conjectures were subsequently proved in a series of significant works in [12–18].

In the above context, it was natural to seek a corresponding holographic characterization for the entanglement negativity of such bipartite states in dual $\text{CFT}_d$s. This was initially attempted for the pure vacuum state of dual $\text{CFT}_d$s in [19]. Subsequently a comprehensive holographic construction for the entanglement negativity of both pure and mixed states in dual $\text{CFT}_{1+1}$s was advanced in the context of the $\text{AdS}_3/\text{CFT}_2$ [20–23] scenario. These proposals were substantiated by a large central charge analysis of the entanglement negativity for $\text{CFT}_{1+1}$s utilizing the monodromy technique in [13, 24–26]. Subsequently, the covariant extension of the holographic entanglement negativity constructions described above were advanced for bipartite states in $\text{CFT}_{1+1}$s dual to non-static $\text{AdS}_3$ backgrounds following the HRT construction [11] in [23,27–29]. Higher dimensional generalizations of the above holographic constructions for bipartite states described by configurations of subsystems with long rectangular strip geometries in $\text{CFT}_d$s dual to bulk static $\text{AdS}_{d+1}$ geometries were proposed in [30–32]. We should mention here that an alternate holographic construction based on the entanglement wedge cross-section [33, 34], for the entanglement negativity of bipartite states in the $\text{AdS}_{d+1}/\text{CFT}_d$ scenario was developed in [35,36]. It has been shown in [37] that this proposal is completely equivalent to the earlier construction for the holographic entanglement negativity upto certain overall multiplicative factors arising from the backreaction of cosmic branes associated with bulk conical defects.[1]

In a separate context, a class of $(1+1)$ dimensional field theories with Galilean conformal symmetries obtained through a parametric İnönü-Wigner contraction of the usual relativistic conformal algebra were investigated in [40–54]. The authors of [43,44] developed a replica technique for computing the entanglement entropy of such Galilean conformal field theories ($\text{GCFT}_{1+1}$). Following this a replica technique to compute the entanglement negativity of bipartite states in a class of such $\text{GCFT}_{1+1}$ was established in [55].

The above class of $\text{GCFT}_{1+1}$s was proposed as possible holographic duals to bulk three-dimensional gravity in asymptotically flat space-times [56] in the framework of flat space holography [57, 58]. The asymptotic symmetry algebra of the bulk geometry was described

---

[1]For more recent developments see [38,39].

by the infinite dimensional Bondi-Metzner-Sachs (BMS$_3$) algebra isomorphic to the Galilean conformal algebra in $1+1$ dimensions (GCA$_2$). The authors of [59] computed the holographic entanglement entropy of a single interval in the corresponding dual BMS$_3$ field theory located at the null infinity of the bulk asymptotically flat geometry. Interestingly in [60], the authors established a holographic construction for the entanglement entropy in the dual BMS$_3$ field theories described above, through a generalization of the covariant HRT construction [11] first proposed in [59]. From a different perspective, the authors of [43] obtained the above flat space holography results utilizing the Chern-Simons formulation of three-dimensional gravity [61] and the Wilson line prescription [62].

The above developments bring the critical issue of a holographic description of mixed state entanglement for these dual GCFT$_{1+1}$ into sharp focus. In this article we address this issue through the BMS$_3$/GCA$_2$ correspondence [58–60]. In this context we establish holographic constructions to compute the entanglement negativity of bipartite states in GCFT$_{1+1}$s dual to bulk asymptotically flat $(2 + 1)$ dimensional Einstein Gravity and Topologically Massive Gravity (TMG) [59, 60, 63–66], following the corresponding constructions for relativistic CFT$_{1+1}$s described in [20, 21, 55]. Interestingly our results match exactly with the universal parts of the corresponding replica technique results obtained in [55]. For the mixed state of disjoint intervals in proximity we substantiate our results through a rigorous geometric monodromy analysis [67] to obtain the corresponding large central charge limit.

This article is organized as follows. In section 2 we briefly recollect the salient features of GCFT$_{1+1}$s and the BMS$_3$/GCA$_2$ correspondence. The replica techniques developed in [43, 44, 55] for computing the entanglement entropy and negativity respectively in such GCFT$_{1+1}$s are reviewed in section 3. In section 4 we describe the covariant construction for computing the entanglement entropy in [59, 60]. In particular, we apply this covariant prescription to obtain the entanglement entropy for a single interval in a GCFT$_{1+1}$ describing a finite-sized system and find perfect agreement with [43, 44]. In section 5, we establish our flat-holographic constructions for computing the entanglement negativity for a single and two adjacent intervals in GCFT$_{1+1}$s dual to Einstein gravity in the bulk asymptotically flat spacetimes. The holographic construction for computing the entanglement negativity for the case of two disjoint intervals along with the large central charge analysis is described in section 6. In section 7 we generalize the above constructions to the case of GCFT$_{1+1}$s dual to bulk geometries described by TMG. The special case of the entanglement negativity in flat chiral gravity is discussed in appendix A. Furthermore, in appendix B we provide details of the geometric monodromy analysis and perform a next to leading order computation to substantiate our results. Finally in appendix C we provide further support to the same through a parametric contraction of the corresponding relativistic CFT$_{1+1}$ results reported in [23, 25]. We conclude in section 8 with a summary of our results and discuss future open issues.

## 2    Review of GCFT$_{1+1}$

In this section we review the basics of $(1 + 1)$ dimensional Galilean conformal field theories (GCFT$_{1+1}$) [40–54]. Interestingly the Galilean conformal algebra (GCA$_2$) may be obtained via an İnönü-Wigner contraction of the usual relativistic conformal algebra in two dimensions:

$$t \to t, \qquad x \to \epsilon x, \tag{1}$$

with $\epsilon \to 0$. This is equivalent to the non-relativistic small velocity limit $v \sim \epsilon$. The Galilean conformal transformations acts on the coordinates as

$$t \to f(t), \quad x \to f'(t)x + g(t), \tag{2}$$

which can be thought of as diffeomorphisms and $t$-dependent shifts, respectively. These are generated by the Nöether charges which, in the plane representation, are given by

$$L_n = t^{n+1}\partial_t + (n+1)t^n x \partial_x, \quad M_n = t^{n+1}\partial_x, \tag{3}$$

which obey the Lie algebra with different central extensions in each sector[2]:

$$\begin{aligned}
[L_n, L_m] &= (m-n)L_{n+m} + \frac{c_L}{12}(n^3-n)\delta_{n+m,0}, \\
[L_n, M_m] &= (m-n)M_{n+m} + \frac{c_M}{12}(n^3-n)\delta_{n+m,0}, \\
[M_n, M_m] &= 0,
\end{aligned} \tag{4}$$

where $c_L$ and $c_M$ are the central charges for the GCA. The cylinder and plane representations are related via the transformation [45,67]

$$x = e^{i\phi}, \qquad t = iu\,e^{i\phi}. \tag{5}$$

The maximally commuting subalgebra is that of the generators $\{L_0, M_0\}$ and the representations are labelled by their eigenvalues (the conformal weights) $h_L$ and $h_M$ in order to construct the highest weight representation.

$$L_0 |h_L, h_M\rangle = h_L |h_L, h_M\rangle, \qquad M_0 |h_L, h_M\rangle = h_M |h_L, h_M\rangle. \tag{6}$$

The two point correlator of primary fields may be written down utilizing the Galilean conformal symmetry as [42,55]

$$\langle V_1(x_1, t_1) V_2(x_2, t_2)\rangle = C^{(2)} \delta_{h_L^1 h_L^2} \delta_{h_M^1 h_M^2} t_{12}^{-2h_L^1} \exp\left(-2h_M^1 \frac{x_{12}}{t_{12}}\right), \tag{7}$$

where $(h_L^1, h_M^1)$ and $(h_L^2, h_M^2)$ are the conformal weights of the primary fields $V_1$ and $V_2$ respectively, $C^{(2)}$ is a normalization constant and $x_{12} = x_1 - x_2$, $t_{12} = t_1 - t_2$. In a similar manner it is easy to determine the three point function of primary fields in a GCFT$_{1+1}$ to be [42,55]

$$\begin{aligned}
\langle V_1(x_1, t_1) V_2(x_2, t_2) V_3(x_3, t_3)\rangle =& C^{(3)} t_{12}^{-(h_L^1+h_L^2-h_L^3)} t_{23}^{-(h_L^2+h_L^3-h_L^1)} t_{13}^{-(h_L^1+h_L^3-h_L^2)} \\
&\times \exp\Big[-(h_M^1+h_M^2-h_M^3)\frac{x_{12}}{t_{12}} - (h_M^2+h_M^3-h_M^1)\frac{x_{23}}{t_{23}} \\
&\quad - (h_M^1+h_M^3-h_M^2)\frac{x_{13}}{t_{13}}\Big],
\end{aligned} \tag{8}$$

where the $V_i$'s are primary fields with weights $\{(h_L^i, h_M^i)\}$ and $x_{ij} = x_i - x_j$, $t_{ij} = t_i - t_j$ with $(i = 1, 2, 3)$ respectively and $C^{(3)}$ is a constant. Similarly, the four-point function of primary fields in the GCFT$_{1+1}$ may be expressed as [55]

$$\begin{aligned}
\left\langle \prod_{i=1}^4 V_i(x_i, t_i)\right\rangle =& \frac{t_{13}^{h_L^1+h_L^3} t_{24}^{h_L^2+h_L^4}}{t_{12}^{h_L^1+h_L^2} t_{23}^{h_L^2+h_L^3} t_{34}^{h_L^3+h_L^4} t_{14}^{h_L^1+h_L^4}} \exp\Big[\frac{x_{13}}{t_{13}}(h_M^1+h_M^3) + \frac{x_{24}}{t_{24}}(h_M^2+h_M^4) \\
&- \frac{x_{12}}{t_{12}}(h_M^1+h_M^2) - \frac{x_{23}}{t_{23}}(h_M^2+h_M^3) - \frac{x_{34}}{t_{34}}(h_M^3+h_M^4) \\
&- \frac{x_{14}}{t_{14}}(h_M^1+h_M^4)\Big] \mathcal{G}(t, \frac{x}{t}),
\end{aligned} \tag{9}$$

---

[2]Note that we are working in the plane representation which differs from the familiar cylinder representation used in [41,42] by a negative sign in the GCA.

where $\{(h_L^i, h_M^i)\}$ are the weights of the primary fields $V_i(x_i, t_i)$ with $(i = 1, 2, 3, 4)$ and

$$t = \frac{t_{12} t_{34}}{t_{13} t_{24}}, \qquad \frac{x}{t} = \frac{x_{12}}{t_{12}} + \frac{x_{34}}{t_{34}} - \frac{x_{13}}{t_{13}} - \frac{x_{24}}{t_{24}}, \tag{10}$$

are the non-relativistic counterparts of the cross ratio $x$ in the relativistic CFT$_{1+1}$s. In eq. (9), $\mathcal{G}(t, \frac{x}{t})$ is a non-universal function of the cross ratios that depends on the full operator content of the specific field theory.

Interestingly the GCFT$_{1+1}$s are equivalent to the BMS$_3$ field theories at the level of the algebra [58]. This leads to a conjectured GCA$_2$/BMS$_3$ correspondence between the asymptotic symmetry algebra of three dimensional Minkowski spacetime at null infinity and the above class of GCFT$_{1+1}$ [44, 45, 58]. Note that the central charges of these contracted algebras are related with the parent Virasoro central charges as [58]

$$c_L = c + \bar{c}, \quad c_M = \epsilon(c - \bar{c}), \tag{11}$$

for GCA$_2$, and as

$$c_L = \epsilon(c - \bar{c}), \quad c_M = c + \bar{c}, \tag{12}$$

for BMS$_3$. Also, the kinematics in the two sectors are related by the replacement $x \longleftrightarrow t$ [44]. We will be using the BMS$_3$/GCA$_2$ correspondence for the computations in the context of flat holographic entanglement in sections 5 to 7.

## 3 Entanglement measures in GCFT$_{1+1}$

In this section we briefly review the replica techniques employed to compute the entanglement entropy and entanglement negativity, in the special class of GCFT$_{1+1}$ described above. As in the case of relativistic CFT$_{1+1}$s [3, 4], the entanglement entropy for a bipartite state in these GCFT$_{1+1}$s may be computed using a replica technique developed in [43, 44]. To this end, one considers $n$-copies of the GCFT$_{1+1}$ plane sewed together along cuts describing the intervals (subsystems) under consideration. The partition function on this replica manifold then computes the Renyi entropy $S_A^{(n)}$ for the boosted interval $A$ [3], in terms of the two-point function of twist fields $\Phi_{\pm n}$ inserted at endpoints $\partial_i A$ of the interval $A$ as

$$(1 - n) S_A^{(n)} = Tr \rho_A^n = \langle \Phi_n(\partial_1 A) \Phi_{-n}(\partial_2 A) \rangle, \tag{13}$$

where the twist fields are primary fields of the GCFT$_{1+1}$ with scaling dimensions

$$\Delta_n = \frac{c_L}{24}\left(n - \frac{1}{n}\right), \quad \chi_n = \frac{c_M}{24}\left(n - \frac{1}{n}\right), \tag{14}$$

and $\rho_A^n$ is the reduced density matrix corresponding to the subsystem $A$. The entanglement entropy for the bipartite state corresponding to the interval $A$ in the GCFT$_{1+1}$ may now be obtained by taking the replica limit $n \to 1$ as

$$S_A = \lim_{n \to 1} S_A^{(n)} = \lim_{n \to 1} \partial_n \langle \Phi_n(\partial_1 A) \Phi_{-n}(\partial_2 A) \rangle. \tag{15}$$

Interestingly it was possible to compute the entanglement negativity for mixed states in relativistic CFT$_{1+1}$s through a related replica technique [5–7]. To define the entanglement

---

[3] Note that in the case of GCFT$_{1+1}$s one cannot consider subsystems at a fixed time slice due to the lack of Lorentz invariance. Therefore one must consider Galilean boosted intervals of the form $A = [(x_1, t_1), (x_2, t_2)]$ [43, 55].

negativity in quantum information theory a tripartite system in a pure state consisting of subsystems $A_1, A_2$ and $B$ is considered. Subsequently the degrees of freedom of the subsystem $B$ are traced over to obtain the reduced density matrix of the mixed state configuration described by $A = A_1 \cup A_2$, as $\rho_A = \text{Tr}_B \rho$, where $\rho$ describes the tripartite state $A \cup B$. The entanglement negativity of the bipartite mixed state described by the reduced density matrix $\rho_A$ is then defined as the trace norm of the partially transposed density matrix $\rho_A^{T_2}$ [1, 5–7]

$$\mathcal{E} = \ln \text{Tr} ||\rho_A^{T_2}||, \tag{16}$$

where the trace norm is defined as the sum of absolute eigenvalues of $\rho_A^{T_2}$. The operation of partial transpose is described as

$$\left\langle e_i^{(1)} e_j^{(2)} | \rho_A^{T_2} | e_k^{(1)} e_l^{(2)} \right\rangle = \left\langle e_i^{(1)} e_l^{(2)} | \rho_A | e_k^{(1)} e_j^{(2)} \right\rangle, \tag{17}$$

where $|e_i^{(1)}\rangle$ and $|e_j^{(2)}\rangle$ are the basis elements for the Hilbert spaces $\mathcal{H}_1$ and $\mathcal{H}_2$ corresponding to $A_1$ and $A_2$, respectively.

Next we briefly discuss the replica construction for computing the entanglement negativity of bipartite states in a $\text{GCFT}_{1+1}$ developed in [55] which closely follows [5, 6] for relativistic $\text{CFT}_{1+1}$.

As for the relativistic $\text{CFT}_{1+1}$, in this case one considers a replicated manifold described by $n_e$-copies (with $n_e$ even) of the $\text{GCFT}_{1+1}$ plane glued together in an appropriate fashion [55]. The entanglement negativity for the bipartite mixed state configuration $A \equiv A_1 \cup A_2$ may then be obtained through a replica technique as

$$\mathcal{E} = \lim_{n_e \to 1} \log \text{Tr}(\rho_A^{T_2})^{n_e}. \tag{18}$$

In eq. (18), we have used the replica limit $n_e \to 1$ and the quantity $\text{Tr}(\rho_A^{T_2})^{n_e}$ can be expressed in terms of a four-point correlator of twist fields $\Phi_{\pm n_e}$ inserted at the endpoints of the intervals as

$$\text{Tr}(\rho_A^{T_2})^{n_e} = \left\langle \Phi_{n_e}(x_1, t_1) \Phi_{-n_e}(x_2, t_2) \Phi_{-n_e}(x_3, t_3) \Phi_{n_e}(x_4, t_4)) \right\rangle. \tag{19}$$

The authors of [55] computed the entanglement negativity for various bipartite pure and mixed state configurations involving a single interval and two adjacent intervals in a $\text{GCFT}_{1+1}$. In the subsequent sections, we will develop holographic constructions to compute the entanglement negativity for such configurations in a $\text{GCFT}_{1+1}$. Furthermore, in section 6 we will describe a geometric monodromy technique to obtain the universal part of the four-point twist correlator in (19) from which it is possible to establish a holographic construction for the entanglement negativity of the mixed state configuration of two disjoint intervals in proximity.

## 4 Entanglement in flat holography

In this section we review the salient features of the covariant construction in [59, 60] for computing entanglement entropy in flat holography in the spirit of the HRT prescription [11] in the usual AdS/CFT scenario. The entanglement entropy of a bipartite state described by a single interval in the $\text{BMS}_3/\text{GCA}_2$ field theory located at the null infinity of the dual asymptotically flat bulk geometry will be given by the length of a bulk extremal geodesic homologous to the interval. We first consider the case of the $\text{BMS}_3/\text{GCA}_2$ field theory dual to bulk asymptotically flat $(2+1)$-dimensional Einstein Gravity for which the Brown-Henneaux symmetry analysis at null infinity leads to the infinite dimensional $\text{BMS}_3/\text{GCA}_2$ algebra. For the appropriate boundary conditions, the general solution to Einstein equations in the Bondi gauge is [59]

$$ds^2 = \Theta(\phi) du^2 - 2\, du\, dr + 2 \left[ \Xi(\phi) + \frac{u}{2} \partial_\phi \Theta(\phi) \right] du\, d\phi + r^2 d\phi^2, \tag{20}$$

where $u = t - r$ in the (retarded) Eddington-Finkelstein time, $r$ is the holographic coordinate, and $\Theta(\phi)$ and $\Xi(\phi)$ are arbitrary functions of the angular coordinate $\phi$. It is interesting to note that by construction the holographic direction is null.

As stated earlier the flat space holographic principle requires a dual $\text{BMS}_3/\text{GCA}_2$ field theory located at the null infinity of the bulk asymptotically flat spacetime. The corresponding central charges for this dual field theory are obtained from the asymptotic symmetry analysis as [59, 67–69]

$$c_L = 0, \quad c_M = \frac{3}{G}. \tag{21}$$

It is interesting to note that the global subalgebra of the $\text{BMS}_3$ group is identical to the Poincare algebra. Therefore the corresponding conformal weights $\Delta$ and $\chi$ which label the representations of the $\text{BMS}_3/\text{GCA}_2$ must correspond to the quadratic Casimirs of the Poincare algebra. This indicates the presence of a massive particle with spin propagating in the bulk geometry. For Einstein gravity in the bulk however the equations (14) and (21) indicate that $\Delta = 0$, which corresponds to the propagation of a spinless massive particle in the bulk spacetime [60].

## 4.1  Holographic entanglement in flat Minkowski space

We start with the holographic computation of the entanglement entropy for a single interval in the vacuum state of a $\text{GCFT}_{1+1}$. To this end we consider the dual geometry of the bulk flat $(2+1)$ dimensional Minkowski spacetime in Eddington-Finkelstein coordinates which is given as

$$ds^2 = dr^2 - du^2 + r^2 d\phi^2, \tag{22}$$

where the coordinates are as described earlier. We consider an interval $A = [(u_1^\partial, \phi_1^\partial), (u_2^\partial, \phi_2^\partial)]$ on the dual $\text{GCFT}_{1+1}$ plane located at the null infinity of the flat spacetime. It was shown in [60] the length of the bulk extremal curve joining the endpoints $\partial_i A$ ($i = 1, 2$) of the interval, is given by

$$L_{\text{tot}}^{\text{extr}} = \left| \frac{u_{12}^\partial}{\tan \frac{\phi_{12}^\partial}{2}} \right|. \tag{23}$$

Note that the bulk extremal curve consists of two null curves descending from the endpoints $\partial_i A$ which do not intersect and a third extremal curve is required to connect them. Recall that for Einstein gravity in the bulk we have $c_L = 0$ from eq. (21). Therefore, as described in [60], in the large $c_M$ limit, the twist fields inserted at the endpoints of the interval correspond to a bulk propagating particle of mass $m_n = \chi_n$. Consequently the two point correlator (13) of these twist fields can be expressed as the exponential of the on-shell action of such a particle propagating along an extremal trajectory $X^\mu(s)$ homologous to the interval. With such an identification we write following [60]:

$$\langle \Phi_n(\partial_1 A) \Phi_{-n}(\partial_2 A) \rangle = e^{-m_n S_{\text{on-shell}}}, \tag{24}$$

where $m_n = \chi_n$ and

$$S_{\text{on-shell}} = \sqrt{\eta_{\mu\nu} \dot{X}^\mu \dot{X}^\nu} = L_{\text{tot}}^{\text{extr}}. \tag{25}$$

Therefore the entanglement entropy for the single interval $A$ in eq. (15) is given by the flat space analog of the HRT formula [59, 60, 70]

$$S_A = \frac{1}{4G} L_{\text{tot}}^{\text{extr}} = \frac{1}{4G} \left| \frac{u_{12}^\partial}{\tan \frac{\phi_{12}^\partial}{2}} \right|, \tag{26}$$

where we have used eq. (21).

## 4.2 Holographic entanglement in global Minkowski orbifolds

Next we focus on a $\mathrm{GCFT}_{1+1}$ compactified on a spatial circle of circumference $L$. The dual geometry is the global Minkowski orbifold, which is described as the quotient of the usual Minkowski spacetime with the compact spatial circle [59]:

$$(u, \phi) \sim (u, \phi + L_\phi). \tag{27}$$

The metric for global Minkowski orbifolds reads [59]

$$ds^2 = -\left(\frac{2\pi}{L_\phi}\right)^2 du^2 - 2du\,dr + r^2 d\phi^2. \tag{28}$$

The holographic entanglement entropy of the boosted interval $A = [(u_1^\partial, \phi_1^\partial), (u_2^\partial, \phi_2^\partial)]$ is obtained from the length of a bulk extremal curve homologous to the interval in the dual field theory. Note that the bulk geodesics are not necessarily straight lines for this case which renders the analysis to be more involved than for the bulk flat Minkowski spacetime. To this end we compute the geodesic length in the Cartesian coordinates and map the endpoints to the global Minkowski orbifold through the the coordinate transformations which implements the quotienting [59, 60]. These coordinate transformations are given as

$$r = \frac{2\pi}{L_\phi}\sqrt{x^2 - t^2},$$
$$u = \left(\frac{L_\phi}{2\pi}\right)^2 \left[\frac{2\pi i}{L_\phi} y - r\right], \tag{29}$$
$$\phi = \frac{L_\phi}{2\pi i}\log\left[\frac{2\pi i}{L_\phi}\frac{(t-x)}{r}\right] = \frac{L_\phi}{2\pi}\sin^{-1}\left[\frac{\pi(t-x)}{L_\phi r} + \frac{L_\phi r}{4\pi(t-x)}\right].$$

Inverting these relations, we obtain

$$x = \frac{L_\phi r}{2\pi}\sin\left(\frac{2\pi\phi}{L_\phi}\right), \quad t = \frac{L_\phi r}{2\pi}\cos\left(\frac{2\pi\phi}{L_\phi}\right), \quad y = \frac{L_\phi}{2\pi i}r - \frac{2\pi i}{L_\phi}u. \tag{30}$$

The length of the bulk geodesic from $y_1$ to $y_2$ obtained through this procedure is expressed as

$$L(y_1, y_2) = \frac{L_\phi}{2\pi}\left[2r_1 r_2\left(1 - \cos\frac{2\pi(\phi_1\phi_2)}{L_\phi}\right) - \frac{8\pi^2}{L_\phi^2}(r_1 - r_2)(u_1 - u_2) - \left(\frac{2\pi}{L_\phi}\right)^4(u_1 - u_2)^2\right]^{1/2}. \tag{31}$$

Similar to the previous case of the bulk pure Minkowski spacetime [60], we have null hypersurfaces on which the null curves descending from the endpoints $(u_i^\partial, \phi_i^\partial)$ of the boundary interval lie:

$$N_i : \quad \frac{2\pi}{L_\phi}(u_i^\partial - u_i) - 2r_i \sin^2\left(\frac{\pi(\phi_i - \phi_i^\partial)}{L_\phi}\right) = 0. \tag{32}$$

The invariant length between $y_i \in N_i$ and the boundary endpoint $\partial_i A$ is given by

$$L(y_i, \partial_i A) = \frac{L_\phi}{2\pi}r_i \sin\left[\frac{2\pi(\phi_i - \phi_i^\partial)}{L_\phi}\right]. \tag{33}$$

The null lines now correspond to $u_i = u_i^\partial$, $\phi_i = \phi_i^\partial$ which usually do not intersect and another extremal curve connecting the null lines is required. The total length of the extremal curve may then be expressed as follows

$$L_{\mathrm{tot}} = L^{\mathrm{extr}}(y_1, \partial_1 A) + L^{\mathrm{extr}}(y_1, y_2) + L^{\mathrm{extr}}(y_2, \partial_2 A) = L^{\mathrm{extr}}(y_1, y_2). \tag{34}$$

The extremization of the length in eq. (31) with respect to the position of the endpoints leads to

$$\frac{\partial L_{\text{tot}}}{\partial r_i} = 0 \implies r_2 = \frac{4\pi^2 u_{12}^{\partial}/L_{\phi}^2}{1 - \cos\left(\frac{2\pi\phi_{12}^{\partial}}{L_{\phi}}\right)} = -r_1 \,. \tag{35}$$

Substituting this back into the expression (34) we obtain the length of the extremal curve homologous to the interval as

$$L_{\text{tot}}^{\text{extr}} = \frac{2\pi u_{12}^{\partial}}{L_{\phi}} \cot\left(\frac{\pi\phi_{12}^{\partial}}{L_{\phi}}\right) \,. \tag{36}$$

Consequently the holographic entanglement entropy for the interval $A$ in the dual field theory is given by

$$S_A = \frac{1}{4G} L_{\text{tot}}^{\text{extr}} = \frac{c_M}{6} \frac{\pi u_{12}^{\partial}}{L_{\phi}} \cot\left(\frac{\pi\phi_{12}^{\partial}}{L_{\phi}}\right) \,, \tag{37}$$

where in the last expression we have used eq. (21). This matches with the $c_L = 0$ part of the entanglement entropy of the single interval in the $\text{BMS}_3/\text{GCA}_2$ field theory dual to the global Minkowski orbifold obtained in [43].

## 4.3 Holographic entanglement in flat space cosmologies

In this subsection we will consider a finite temperature $\text{GCFT}_{1+1}$ with a compactified thermal cycle $(u, \phi) \sim (u + i\beta_u, \phi + i\beta_{\phi})$. The corresponding holographic dual is another interesting quotient of Minkowski spacetime called Flat Space Cosmology (FSC), with the metric [43–45]

$$ds^2 = M du^2 - 2\,du\,dr + J\,du\,d\phi + r^2 d\phi^2 \,, \tag{38}$$

where the temperatures in the dual field theory at null infinity are related to the ADM mass and angular momentum of the spacetime as $\beta_u = \pi J M^{-3/2}$ and $\beta_{\phi} = 2\pi M^{-1/2}$. For this geometry a similar computation of the geodesic length as above yields the following expression for the geodesic length [60]

$$L_{\text{tot}}^{\text{extr}} = \sqrt{M}\left(u_{12}^{\partial} + \frac{J}{2M}\phi_{12}^{\partial}\right)\coth\left(\frac{\sqrt{M}\phi_{12}^{\partial}}{2}\right) - \frac{J}{M} \,. \tag{39}$$

We are mainly interested in the non-rotating geometry, therefore putting $J = 0$ and writing $\beta$ for $\beta_{\phi}$, we obtain

$$L_{\text{tot}}^{\text{extr}} = \frac{2\pi u_{12}^{\partial}}{\beta} \coth\left(\frac{\pi\phi_{12}^{\partial}}{\beta}\right) \,, \tag{40}$$

and consequently the holographic entanglement entropy for the boundary interval $A$ in the thermal $\text{GCFT}_{1+1}$ is given by

$$S_A = \frac{c_M}{6} \frac{\pi u_{12}^{\partial}}{\beta} \coth\left(\frac{\pi\phi_{12}^{\partial}}{\beta}\right) \,. \tag{41}$$

# 5 Holographic entanglement negativity in flat Einstein gravity

In this section we detail the holographic constructions for computing the entanglement negativity of bipartite states in the class of $\text{GCFT}_{1+1}$s dual to bulk asymptotically flat geometries

using results from the flat space holography described in the last section 4. In particular we will consider the asymptotically flat bulk spacetimes described by Einstein gravity for which the asymptotic symmetry analysis reveals that the dual GCFT$_{1+1}$s possess only one non zero central charge $c_M$ (cf eq. (21)). We will first describe the holographic construction to compute the entanglement negativity of various bipartite states described by a single interval in the dual GCFT$_{1+1}$. These include a single interval for a GCFT$_{1+1}$ in its ground state, a GCFT$_{1+1}$ describing a finite-sized system and a GCFT$_{1+1}$ at a finite temperature respectively. Next we turn our attention to the configuration of two adjacent intervals in the dual GCFT$_{1+1}$ and establish holographic constructions to compute the entanglement negativity for the configurations described above using the results of flat space holography. The case of the two disjoint intervals will require an analysis of the semi-classical Galilean conformal blocks in the large central charge limit of the GCFT$_{1+1}$. We will postpone the discussion of such configurations till section 6.

## 5.1 Holographic entanglement negativity for a single interval

In this subsection we will consider various bipartite pure and mixed states consisting of a single interval in a large system described by a GCFT$_{1+1}$. We start with the simplest configurations of bipartite pure states described by a single interval $A \equiv [(x_1, t_1), (x_2, t_2)]$. As described in [55], the corresponding entanglement negativity involves a two-point correlator of composite twist fields, given by

$$\mathcal{E} = \lim_{n_e \to 1} \log \left\langle \Phi_{n_e}^2(x_1, t_1) \Phi_{-n_e}^2(x_2, t_2) \right\rangle. \tag{42}$$

We now apply the flat space holographic dictionary in eqs. (24) and (25) to obtain the following form for the above twist correlator:

$$\left\langle \Phi_{n_e}^2(x_1, t_1) \Phi_{-n_e}^2(x_2, t_2) \right\rangle = \left( \left\langle \Phi_{n_e/2}(x_1, t_1) \Phi_{-n_e/2}(x_2, t_2) \right\rangle \right)^2 = e^{-2 \chi_{n_e/2} L_{12}^{\text{extr}}}, \tag{43}$$

where $\chi_{n_e/2}$ is the non-trivial scaling dimension of the twist fields $\Phi_{\pm n_e/2}$ and $L_{12}^{\text{extr}}$ is the length of the bulk extremal curve homologous to the interval in question. In obtaining eq. (43), we have made use of the fact that for pure states the two point correlator of composite twist operators factorizes into that of usual twist operators spanning half of the replica geometry [55]. From eq. (14), in the replica limit $n_e \to 1$, we have $\chi_{n_e/2} \to -\frac{c_M}{16}$[4], and therefore we obtain the following expression for the entanglement negativity of a pure state described by a single interval $A$ in a holographic GCFT$_{1+1}$:

$$\mathcal{E} = \frac{3}{8G} L_A, \tag{44}$$

where we have made use of eq. (21). In the following, we will employ our holographic proposal in eq. (44) to compute the holographic entanglement negativity in some pure quantum states in a holographic GCFT$_{1+1}$. Particularly we will investigate the case of a single interval in the ground state of the GCFT$_{1+1}$, which is dual to the asymptotically flat pure Minkowski spacetime. Then we will turn our attention to the pure state described by the single interval in a finite-sized system described by a GCFT$_{1+1}$ compactified on a spatial cylinder, which is dual to the boost orbifold of Minkowski spacetime. We will find that the results obtained using our holographic formula will reproduce the universal behaviour of the entanglement negativity for both of these configurations [55]. Later, in subsection 5.1.3 we will consider the mixed state configuration of a single interval at a finite temperature which involves a particular four-point twist correlator in the large central charge limit.

---

[4] Note that the negative scaling dimension of the twist fields $\Phi_{n_e}^2$ and $\Phi_{n_e/2}$ in the replica limit $n_e \to 1$ has to be understood only in the sense of an analytic continuation.

### 5.1.1 Single interval at zero temperature

To obtain the entanglement negativity in the bipartite pure state configuration described by a single boosted interval in a GCFT$_{1+1}$ (cf. footnote 3) at zero temperature we use the results from the flat space holography reviewed in section 4 . At this point, we recall that the computation of the length of the extremal geodesic in the dual gravity theory in cylindrical coordinates $(u, \phi)$ results in eq. (23) [60]. In the planar coordinates in eq. (5) [45, 60] this translates to

$$L_{12}^{extr} = 2\frac{x_{12}}{t_{12}}. \tag{45}$$

Therefore, using the above expression for $L_{12}^{extr}$, we obtain the entanglement negativity for a single interval in a GCFT$_{1+1}$ at zero temperature from eq. (44) to be

$$\mathcal{E} = \frac{3}{8G}L_A = \frac{c_M}{4}\frac{x_{12}}{t_{12}}. \tag{46}$$

This is precisely the result obtained in [55] using field theory methods, for $c_L = 0$. It is interesting to note that we may recast the above expression for entanglement negativity in the form

$$\mathcal{E} = \frac{3}{2}S_A, \tag{47}$$

using the flat space analogue of the HRT formula in eq. (26), where $S_A$ is the entanglement entropy for the single interval $A$ in the GCFT$_{1+1}$ vacuum. This indicates that for pure states the holographic entanglement negativity is given by the Rënyi entropy of order half as in the case of quantum information theory [6].

### 5.1.2 Single interval in a finite-sized system

Next we turn our attention to the computation of holographic entanglement negativity for the pure state configuration of a single boosted interval in a finite-sized system admitting periodic boundary conditions described by a GCFT$_{1+1}$ defined on an infinite cylinder with circumference $L_\phi$. The bulk gravity dual is the global Minkowski orbifold described by the metric in eq. (28). The extremal geodesic length was computed in section 4 and is given by

$$L_{ij}^{\text{extr}} = \frac{2\pi u_{ij}}{L_\phi}\cot\left(\frac{\pi\phi_{ij}}{L_\phi}\right), \tag{48}$$

where $u_{ij} = u_i - u_j$ and $\phi_{ij} = \phi_i - \phi_j$ are the differences in the coordinates of the endpoints of the boundary interval.

We may now employ our holographic proposal in eq. (44) to compute the holographic entanglement negativity for the single boosted interval in a finite-sized system. Utilizing eq. (48) we obtain

$$\mathcal{E} = \frac{c_M}{4}\frac{\pi u_{12}}{L_\phi}\cot\left(\frac{\pi\phi_{12}}{L_\phi}\right), \tag{49}$$

which matches exactly with the universal part of the dual field theory result for $c_L = 0$ [55]. Again using the flat holographic HRT formula in (26) we may express the above result in the form (47).

### 5.1.3 Single interval at a finite temperature

The mixed state configuration described by a single interval in a finite temperature GCFT$_{1+1}$ requires a more careful analysis. To start with we recall that a GCFT$_{1+1}$ at a finite temperature is defined on an infinite cylinder of circumference equal to the inverse temperature $\beta$. The

corresponding entanglement negativity involves a four-point twist correlator on the infinite cylinder arising from the configuration of a single interval sandwiched between two adjacent large but finite intervals [55]. The entanglement negativity may then be obtained through a bipartite limit subsequent to the replica limit. Therefore in order to understand the configuration described by a single interval at a finite temperature, we first consider a four-point twist correlator on the GCFT$_{1+1}$ plane [55] (cf. eq. (9)):

$$
\left\langle \Phi_{n_e}(x_1,t_1)\Phi^2_{-n_e}(x_2,t_2)\Phi^2_{n_e}(x_3,t_3)\Phi_{-n_e}(x_4,t_4) \right\rangle = \frac{k_{n_e}\,k^2_{n_e/2}}{t_{14}^{2\Delta_{n_e}}\,t_{23}^{2\Delta^{(2)}_{n_e}}}\frac{\mathcal{F}_{n_e}(t,x/t)}{t^{\Delta^{(2)}_{n_e}}}
$$
$$
\times \exp\left[-2\chi_{n_e}\frac{x_{14}}{t_{14}} - 2\chi^{(2)}_{n_e}\frac{x_{23}}{t_{23}} - \chi^{(2)}_{n_e}\frac{x}{t}\right],
\tag{50}
$$

where $k_{n_e}$ is a constant that depends on the full operator content of the theory. The corresponding weights of the twist fields $\Phi_{\pm n_e}$ are given in eq. (14), from which one can determine the weights of the composite twist fields $\Phi^2_{\pm n_e}$ as [55]:

$$
\Delta^{(2)}_{n_e} = 2\Delta_{n_e/2} = \frac{c_L}{12}\left(\frac{n_e}{2} - \frac{2}{n_e}\right), \quad \chi^{(2)}_{n_e} = 2\chi_{n_e/2} = \frac{c_M}{12}\left(\frac{n_e}{2} - \frac{2}{n_e}\right).
\tag{51}
$$

Equipped with eq. (7) for the two-point twist correlators, the universal part of the four-point function (which is dominant in the large central charge limit of the GCFT$_{1+1}$) in eq. (50) can be factorized as

$$
\left\langle \Phi_{n_e}(x_1,t_1)\Phi^2_{-n_e}(x_2,t_2)\Phi^2_{n_e}(x_3,t_3)\Phi_{-n_e}(x_4,t_4) \right\rangle
$$
$$
= \left(\left\langle \Phi_{n_e/2}(x_2,t_2)\Phi_{-n_e/2}(x_3,t_3)\right\rangle\right)^2 \left\langle \Phi_{n_e}(x_1,t_1)\Phi_{-n_e}(x_4,t_4)\right\rangle
$$
$$
\times \frac{\left\langle \Phi_{n_e/2}(x_1,t_1)\Phi_{-n_e/2}(x_2,t_2)\right\rangle\left\langle \Phi_{n_e/2}(x_3,t_3)\Phi_{-n_e/2}(x_4,t_4)\right\rangle}{\left\langle \Phi_{n_e/2}(x_1,t_1)\Phi_{-n_e/2}(x_3,t_3)\right\rangle\left\langle \Phi_{n_e/2}(x_2,t_2)\Phi_{-n_e/2}(x_4,t_4)\right\rangle} + \mathcal{O}\left(\frac{1}{c}\right).
\tag{52}
$$

Note that the arbitrary non-universal function of the GCFT$_{1+1}$ cross ratios $\mathcal{F}_{n_e}(t,x/t)$ has been neglected in the above factorization. We may justify this as follows. In the semi-classical limit ($G \to 0$) of the bulk asymptotically flat gravity, the flat space holographic dictionary described in section 4 dictates that the dual GCFT$_{1+1}$ theory has a large central charge $c_M \to \infty$ (cf. eq. (21)). Hence, we require a large central charge analysis of the twist-correlator in eq. (50) for the entanglement negativity before giving its holographic description. In section 6 we will develop a monodromy technique to understand the large central charge behaviour of a specific four-point function of twist fields relevant to the computation of entanglement negativity for the mixed state configuration of two disjoint intervals. There we will show that in the large central charge limit $c_M \to \infty$ the non-universal part of the four-point twist correlator is subleading in comparison to the universal part. In the present context, we assume that the four-point twist correlator in (50) has a similar large-$c_M$ structure and therefore the subleading contributions from the non-universal function $\mathcal{F}_{n_e}(t,x/t)$ in eq. (50) is neglected as shown by the $\mathcal{O}(1/c)$ contribution in eq. (52).

Now we utilize the flat space holographic dictionary in eqs. (24) and (25) to find that the four-point function in eq. (52) may be written in the following form

$$
\left\langle \Phi_{n_e}(x_1,t_1)\Phi^2_{-n_e}(x_2,t_2)\Phi^2_{n_e}(x_3,t_3)\Phi_{-n_e}(x_4,t_4) \right\rangle
$$
$$
= \exp\left[-\chi_{n_e}\,\mathrm{L}^{\text{extr}}_{14} - \chi_{n_e/2}\left(2\mathrm{L}^{\text{extr}}_{23} + \mathrm{L}^{\text{extr}}_{12} + \mathrm{L}^{\text{extr}}_{34} - \mathrm{L}^{\text{extr}}_{13} - \mathrm{L}^{\text{extr}}_{24}\right)\right],
\tag{53}
$$

where $L^{\text{extr}}_{ij}$ denotes the length of the extremal geodesic in the bulk, which connects the points $(x_i,t_i)$ and $(x_j,t_j)$ on the boundary. Figure 1 shows the schematics for the configuration

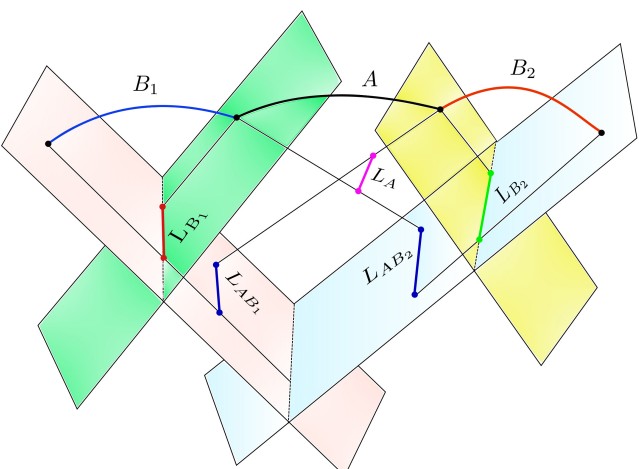

Figure 1: Schematics of the extremal geodesics anchored on different subsystems corresponding to the computation of entanglement negativity for a single interval in a finite temperature GCFT$_{1+1}$. The null planes descending from the boundary endpoints are shown. The non-trivial contributions to the geodesic lengths land on the crossings of the corresponding null planes.

of a single interval $A = [(x_2, t_2), (x_3, t_3)]$ sandwiched between two large auxiliary intervals $B_1 = [(x_1, t_1), (x_2, t_2)]$ and $B_2 = [(x_3, t_3), (x_4, t_4)]$ with $B_1 \cup B_2 \equiv B$. As briefly alluded to in section 4 the orientations of extremal geodesics anchored on different subsystems follow the construction in [60].

From fig. 1 we identify that

$$
\begin{aligned}
L_{12}^{\text{extr}} = L_{B_1}, && L_{23}^{\text{extr}} = L_A, && L_{34}^{\text{extr}} = L_{B_2}, \\
L_{13}^{\text{extr}} = L_{A \cup B_1}, && L_{24}^{\text{extr}} = L_{A \cup B_2}, && L_{14}^{\text{extr}} = L_{A \cup B}.
\end{aligned}
\tag{54}
$$

In the replica limit $n_e \to 1$, we have from eq. (14) $\chi_{n_e} \to 0$ and $\chi_{\frac{n_e}{2}} \to -\frac{c_M}{16}$. Therefore, eq. (53) leads to the following expression for the holographic entanglement negativity

$$
\mathcal{E} = \lim_{B \to A^c} \frac{3}{16G} \left( 2L_A + L_{B_1} + L_{B_2} - L_{A \cup B_1} - L_{A \cup B_2} \right).
\tag{55}
$$

In writing eq. (55) from eq. (53) we have first taken the replica limit $n_e \to 1$ and subsequently taken the bipartite limit $B \to A^c$ in which the intervals $B_1$ and $B_2$ are extended to infinity such that $B_1 \cup B_2 = A^c$ [55]. We have also utilized the fact that for Einstein gravity the asymptotic symmetry analysis following the Brown-Henneaux procedure [71] dictates that the central charges of the dual GCFT$_{1+1}$ are given by (21). Therefore we conclude that the holographic formula for the entanglement negativity of a single interval in a finite temperature dual GCFT$_{1+1}$ relies on a specific linear combination of the lengths of bulk extremal surfaces homologous to the boundary intervals, as shown in fig. 1. Remarkably the flat-holographic proposal for the entanglement negativity for asymptotically flat gravity in eq. (55) has exactly the same structure as in the AdS/CFT scenario obtained in [20]. Interestingly, implementing the flat-holographic counterpart of the HRT formula in eq. (26) we may rewrite our proposal in eq. (55) in the following form

$$
\begin{aligned}
\mathcal{E} &= \lim_{B \to A^c} \frac{3}{4} \left( 2S_A + S_{B_1} + S_{B_2} - S_{A \cup B_1} - S_{A \cup B_2} \right) \\
&= \lim_{B \to A^c} \frac{3}{4} \left( \mathcal{I}(A; B_1) + \mathcal{I}(A; B_2) \right),
\end{aligned}
\tag{56}
$$

which shows a particular connection between two different entanglement measures, namely the entanglement negativity and the mutual information, in holographic theories. Note however that these measures are quite distinct in the quantum information theory. It is important to mention here that this specific relation in eq. (56) seems to be unique to the configurations described by single intervals in holographic GCFT$_{1+1}$s at a finite temperature.

We now perform an explicit holographic computation of the entanglement negativity for the finite temperature mixed state configuration described by a single Galilean boosted interval in a thermal GCFT$_{1+1}$, using our proposal in eq. (55). The finite temperature field theory is dual to the Minkowski orbifold describing the locally flat geometry of Flat Space Cosmologies (FSC). The length of the extremal geodesic in the FSC geometry with the metric in eq. (38) is given in eq. (39). To relate with the field theory computations in [55] we will consider the non-rotating geometry with $J = 0$[5]. In this non-rotating limit, we obtain another Minkowski orbifold, namely the boosted null orbifold. In this case, the expression for the length of the extremal geodesic homologous to the interval at the boundary in eq. (39) simplifies to eq. (40), namely

$$L_{ij}^{extr} = \sqrt{M}\, u_{12} \coth\left(\frac{\sqrt{M}\,\phi_{ij}}{2}\right) = \frac{2\pi u_{ij}}{\beta} \coth\left(\frac{\pi\phi_{ij}}{\beta}\right), \tag{57}$$

where we have simply written $\beta$ for $\beta_\phi = 2\pi M^{-1/2}$ and $u_{ij} = u_i - u_j$ and $\phi_{ij} = \phi_i - \phi_j$ are the differences in the coordinates of the endpoints of the interval at the boundary. Now substituting for the extremal geodesic length in eq. (55) the holographic entanglement negativity for a single interval in a GCFT$_{1+1}$ at a finite temperature is obtained as

$$\mathcal{E} = \frac{c_M}{4}\left[\frac{\pi u_{12}}{\beta}\coth\left(\frac{\pi\phi_{12}}{\beta}\right) - \frac{\pi u_{12}}{\beta}\right]. \tag{58}$$

In obtaining eq. (58) we have used the understanding that $B \to A^c$ corresponds to taking the lengths of $B_1$ and $B_2$ to infinity. This matches exactly with the $c_L = 0$ version of the universal part of the result obtained from the dual field theory in [55]. Although this stands as a strong consistency check for our proposal, it is important to mention that the analysis leading to eq. (55) relies on the large central charge behaviour of the dual GCFT$_{1+1}$ and a bulk proof remains an open issue.

Finally, it is interesting to note that using the flat space analogue of the HRT formula (26), the expression for the holographic entanglement negativity for a single interval in a GCFT$_{1+1}$ at a finite temperature obtained in eq. (58) can be rewritten in the following form

$$\mathcal{E} = \frac{3}{2}\left(S_A - S^{th}\right), \tag{59}$$

where $S_A$ and $S^{th}$ are the entanglement entropy and the thermal entropy respectively, for the single interval $A$ in the holographic GCFT$_{1+1}$.

## 5.2 Holographic entanglement negativity for adjacent intervals

Having computed the holographic entanglement negativity for various bipartite mixed states involving a single interval in the dual GCFT$_{1+1}$, we now proceed to advance a similar holographic construction for the bipartite states described by two adjacent intervals in a holographic GCFT$_{1+1}$. As described before, the large central charge behaviour for the entanglement negativity in a GCFT$_{1+1}$ indicates the plausibility of a holographic characterization for

---

[5]Note that the FSC geometry is defined for non-vanishing angular momentum $J$. Switching off the angular momentum leads to a Big-Bang like naked singularity [45]. The limit of $J \to 0$ has to be understood in the sense of an analytic continuation.

the entanglement negativity in a dual asymptotically flat spacetime through flat space holography. To this end, we consider two Galilean boosted adjacent intervals $A = [(x_1, t_1), (x_2, t_2)]$ and $B = [(x_2, t_2), (x_3, t_3)]$, as depicted in fig. 2, where the system $A \cup B$ is in a mixed state. We start with the following three-point twist correlator on the GCFT$_{1+1}$ plane relevant to the computation of the entanglement negativity of two adjacent intervals [55] (cf. eq. (8)):

$$\left\langle \Phi_{n_e}(x_1, t_1)\Phi^2_{-n_e}(x_2, t_2)\Phi_{n_e}(x_3, t_3) \right\rangle = k^2_{n_e} K_{\Phi_{n_e}\Phi^2_{-n_e}\Phi_{n_e}} t^{-\Delta^{(2)}_{n_e}}_{12} t^{-\Delta^{(2)}_{n_e}}_{23} t^{-(2\Delta_{n_e}-\Delta^{(2)}_{n_e})}_{13}$$
$$\times \exp\left[ -\chi^{(2)}_{n_e}\frac{x_{12}}{t_{12}} - \chi^{(2)}_{n_e}\frac{x_{23}}{t_{23}} - (2\chi_{n_e}-\chi^{(2)}_{n_e})\frac{x_{13}}{t_{13}} \right]. \quad (60)$$

Utilizing equations (43) and (51) the three-point twist correlator in eq. (60) can be rewritten in the following form

$$\left\langle \Phi_{n_e}(x_1, t_1)\Phi^2_{-n_e}(x_2, t_2)\Phi_{n_e}(x_3, t_3) \right\rangle$$
$$= \mathcal{K}\left\langle \Phi_{n_e}(x_1, t_1)\Phi_{-n_e}(x_3, t_3) \right\rangle \left( \frac{\left\langle \Phi^2_{n_e}(x_1, t_1)\Phi^2_{-n_e}(x_2, t_2) \right\rangle \left\langle \Phi^2_{n_e}(x_2, t_2)\Phi^2_{-n_e}(x_3, t_3) \right\rangle}{\left\langle \Phi^2_{n_e}(x_1, t_1)\Phi^2_{-n_e}(x_3, t_3) \right\rangle} \right)^{1/2}, \quad (61)$$

where the constant $\mathcal{K}$ is given by

$$\mathcal{K} = k^2_{n_e} K_{\Phi_{n_e}\Phi^2_{-n_e}\Phi_{n_e}} k^{(1)} \sqrt{k^{(2)}}. \quad (62)$$

Now using the relation (cf. eq.(43))

$$\left\langle \Phi^2_{n_e}(x_1, t_1)\Phi^2_{-n_e}(x_2, t_2) \right\rangle = \left( \left\langle \Phi_{n_e/2}(x_1, t_1)\Phi_{-n_e/2}(x_2, t_2) \right\rangle \right)^2, \quad (63)$$

the universal part (which gives the dominant contribution to the entanglement negativity in the large-$c_M$ limit) of the three-point twist correlator may be written as

$$\left\langle \Phi_{n_e}(x_1, t_1)\Phi^2_{-n_e}(x_2, t_2)\Phi_{n_e}(x_3, t_3) \right\rangle$$
$$= \mathcal{K}\left\langle \Phi_{n_e}(x_1, t_1)\Phi_{-n_e}(x_3, t_3) \right\rangle \frac{\left\langle \Phi_{n_e/2}(x_1, t_1)\Phi_{-n_e/2}(x_2, t_2) \right\rangle \left\langle \Phi_{n_e/2}(x_2, t_2)\Phi_{-n_e/2}(x_3, t_3) \right\rangle}{\left\langle \Phi_{n_e/2}(x_1, t_1)\Phi_{-n_e/2}(x_3, t_3) \right\rangle}. \quad (64)$$

Finally using the flat holographic dictionary in eqs. (24) and (25), we obtain the universal part of the three-point twist correlator as

$$\left\langle \Phi_{n_e}(x_1, t_1)\Phi^2_{-n_e}(x_2, t_2)\Phi_{n_e}(x_3, t_3) \right\rangle = \exp\left[ -\chi_{n_e} L^{\text{extr}}_{13} - \chi_{n_e/2}\left( L^{\text{extr}}_{12} + L^{\text{extr}}_{23} - L^{\text{extr}}_{13} \right) \right], \quad (65)$$

where $L^{\text{extr}}_{ij}$ denotes the length of the extremal curve connecting the endpoints $(x_i, t_i)$ and $(x_j, t_j)$ of an interval on the boundary. In figure 2, we show the schematics of the extremal curves anchored on the subsystems $A$, $B$ and $A \cup B$ respectively, where we have identified

$$L^{\text{extr}}_{12} = L_A, \quad L^{\text{extr}}_{23} = L_B, \quad L^{\text{extr}}_{13} = L_{A \cup B}. \quad (66)$$

In the replica limit $n_e \to 1$, from eq. (14) we obtain $\chi_{n_e} \to 0$ and $\chi_{\frac{n_e}{2}} \to -\frac{c_M}{16}$ (cf. footnote 4). Note that the large central charge limit has to be taken prior to the replica limit. This order of limits is critical since the scaling dimension of the twist field $\Phi_{n_e}$ vanishes in the replica limit and has to be understood in the sense of an analytic continuation. Hence, eq. (65) leads to the following expression for the holographic entanglement negativity for adjacent intervals

$$\mathcal{E} = \frac{3}{16G}\left( L^{\text{extr}}_{12} + L^{\text{extr}}_{23} - L^{\text{extr}}_{13} \right), \quad (67)$$

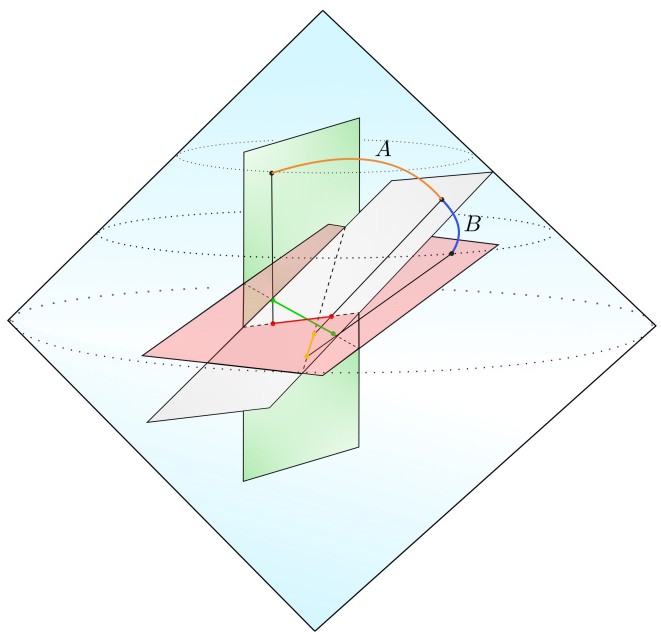

Figure 2: Holographic construction for the computation of the entanglement negativity for two Galilean boosted adjacent intervals $A = (x_1, t_1)$ and $B = (x_2, t_2)$. Extremal geodesics anchored on different subsystems are shown in: red - $L_{12}^{extr} \equiv L_A^{extr}$, yellow - $L_{23}^{extr} \equiv L_B^{extr}$, green- $L_{13}^{extr} \equiv L_{A \cup B}^{extr}$

where we have again used the fact that for Einstein gravity the central charges of the dual $GCFT_{1+1}$ are given by eq. (21). Therefore we conclude that the flat holographic entanglement negativity for two adjacent intervals in the class of holographic $GCFT_{1+1}$s that we consider in the present article, is expressed in terms of a specific algebraic sum of the lengths of bulk extremal geodesics anchored on the endpoints of the intervals at the boundary. Remarkably the flat space holographic formula in eq. (67) has exactly the same structure as its relativistic counterpart obtained in [21].

It is interesting to note that the holographic entanglement negativity formula in eq. (67) may be recast, using the flat holographic HRT formula of [60] in eq. (25), in the form of another entanglement measure in such holographic $GCFT_{1+1}$s, namely the mutual information:

$$\mathcal{E} = \frac{3}{4}(S_A + S_B - S_{A \cup B}) = \frac{3}{4}\mathcal{I}(A:B).\tag{68}$$

Note that this particular connection between the two different entanglement measures is special to the configuration of two adjacent intervals in holographic $GCFT_{1+1}$s.

### 5.2.1 Adjacent intervals at zero temperature

We start with the mixed state configuration of two adjacent intervals in the vacuum state of the boundary $GCFT_{1+1}$ for which the bulk dual geometry is that of Minkowski spacetime. Substituting eq. (45) for the length of the extremal geodesic in pure Minkowski spacetime dual to the $GCFT_{1+1}$ vacuum, in the expression (67) for the holographic entanglement negativity for adjacent intervals, we obtain

$$\mathcal{E} = \frac{c_M}{8}\left(\frac{x_{12}}{t_{12}} + \frac{x_{23}}{t_{23}} - \frac{x_{13}}{t_{13}}\right).\tag{69}$$

This matches exactly with the dual field theory result for $c_L = 0$ in [55].

### 5.2.2 Adjacent intervals at a finite temperature

Next we turn our attention to the holographic computation of the entanglement negativity for the bipartite mixed state configuration of two adjacent intervals in a thermal GCFT$_{1+1}$ defined on an infinite cylinder compactified in the timelike direction. The corresponding bulk dual is the $J = 0$ FSC geometry described in section 4. Substituting eq. (57) for the length of the extremal geodesic, in eq. (67), we obtain

$$\mathcal{E} = \frac{c_M}{8} \left[ \frac{\pi u_{12}}{\beta} \coth\left( \frac{\pi \phi_{12}}{\beta} \right) + \frac{\pi u_{23}}{\beta} \coth\left( \frac{\pi \phi_{23}}{\beta} \right) - \frac{\pi u_{13}}{\beta} \coth\left( \frac{\pi \phi_{13}}{\beta} \right) \right]. \tag{70}$$

Again this matches exactly with the dual field theory result for $c_L = 0$ in [55].

### 5.2.3 Adjacent intervals in a finite-sized system

Finally we compute the holographic entanglement negativity for the bipartite mixed state configuration of two adjacent intervals in a finite-sized system described by a GCFT$_{1+1}$ with periodic boundary conditions defined on a spatially compactified cylinder. The bulk dual is the global Minkowski orbifold in eq. (28) described in section 4. Utilizing the length for extremal geodesics given in eq. (36), we obtain from eq. (67)

$$\mathcal{E} = \frac{c_M}{8} \left[ \frac{\pi u_{12}}{L_\phi} \cot\left( \frac{\pi \phi_{12}}{L_\phi} \right) + \frac{\pi u_{23}}{L_\phi} \cot\left( \frac{\pi \phi_{23}}{L_\phi} \right) - \frac{\pi u_{13}}{L_\phi} \cot\left( \frac{\pi \phi_{13}}{L_\phi} \right) \right], \tag{71}$$

which is exactly the result in [55] obtained from the dual field theory computations, for $c_L = 0$.

## 6 Holographic entanglement negativity for two disjoint intervals

In this section we proceed to establish a holographic conjecture for computing the entanglement negativity in the context of flat space holography for the bipartite mixed state configuration of two disjoint intervals in the dual GCFT$_{1+1}$. As briefly alluded to in subsection 5.1.3, the computation of the entanglement negativity for such configurations involves the large central charge analysis of a particular four-point twist correlator. From eq. (9), it is clear that the GCFT$_{1+1}$ four-point function involves an arbitrary function of the cross ratios which depends on the full operator content of the specific field theory under consideration. Also, for Einstein gravity in the bulk the semi-classical limit in the gravitational theory ($G \to 0$) corresponds to the large central charge limit $c_M \to \infty$ in the dual GCFT$_{1+1}$. Motivated by these considerations, in the following we advance a holographic proposal for computing the entanglement negativity for two disjoint intervals in a GCFT$_{1+1}$.

Before proceeding, we briefly review the computation of entanglement negativity for two disjoint intervals in the AdS$_3$/CFT$_2$ scenario performed in [23]. In [25], the authors demonstrated that the entanglement negativity for two disjoint intervals in a CFT$_2$ vanishes in the s-channel ($x \to 0$) where the two intervals are far away, while remains non-trivial in the t-channel ($x \to 1$) which corresponds to the two intervals being in close proximity. Inspired by these findings, the authors in [23] performed a monodromy analysis of the semi-classical structure of the following four-point function in the vacuum state of a generic CFT$_2$:

$$\left\langle \mathcal{T}_{n_e}(z_1) \bar{\mathcal{T}}_{n_e}(z_2) \bar{\mathcal{T}}_{n_e}(z_3) \mathcal{T}_{n_e}(z_4) \right\rangle = z_{13}^{-2\Delta_{n_e}} z_{24}^{-2\Delta_{n_e}} x^{-2\Delta_{n_e}} \mathcal{G}_{n_e}(x), \quad x = \frac{z_{12} z_{34}}{z_{13} z_{24}}, \tag{72}$$

where $\mathcal{T}_{n_e}$ and $\bar{\mathcal{T}}_{n_e}$ are respectively the twist and anti-twist fields inserted at the endpoints of the two disjoint intervals $[z_1, z_2]$ and $[z_3, z_4]$. In eq. (72), $x$ is the usual $CFT_2$ cross ratio and $\mathcal{G}_{n_e}(x)$ is an arbitrary function of cross ratio. Subsequently, it was found in [23] that the entanglement negativity for the two disjoint intervals in proximity obtained through this procedure has a holographic description in terms of a particular linear combination of the lengths of bulk spacelike geodesics homologous to specific subsystems.

In the following we will utilize similar semi-classical techniques developed in [67] to compute the entanglement negativity for two disjoint intervals $A_1 = [(x_1, t_1), (x_2, t_2)]$ and $A_2 = [(x_3, t_3), (x_4, t_4)]$. This involves an analysis of the large-central charge behaviour of the following four-point twist-correlator in a $GCFT_{1+1}$ vacuum [6]:

$$\left\langle \Phi_{n_e}(X_1) \Phi_{-n_e}(X_2) \Phi_{-n_e}(X_3) \Phi_{n_e}(X_4) \right\rangle = t_{23}^{-2\Delta_{n_e}} t_{14}^{-2\Delta_{n_e}} t^{-2\Delta_{n_e}}$$
$$\times \exp\left[-2\chi_{n_e}\frac{x_{23}}{t_{23}} - 2\chi_{n_e}\frac{x_{14}}{t_{14}} - 2\chi_{n_e}\frac{x}{t}\right] \mathcal{F}(t, \frac{x}{t}). \tag{73}$$

In eq. (73), $t, x/t$ are the non-relativistic cross ratios given in eq. (10) and $\mathcal{F}(t, \frac{x}{t})$ is a non-universal function of cross ratios that depends on the specific operator content of the field theory. In particular, we will focus only on the behaviour of the four-point twist correlator in eq. (73) in the $t$-channel defined as $t \to 1, x \to 0$ [7], which renders the two disjoint intervals in close proximity. We will be working with the $GCFT_{1+1}$s with only one non-vanishing central charge $c_M$ for which the dual bulk geometry is described by Einstein gravity.

## 6.1 Four-point twist correlator at Large $c_M$

In this subsection we explicitly compute the large central charge limit $c_M \to \infty$ of the Galilean conformal block corresponding to the four-point function in eq. (73). To proceed, we recall some salient features of $GCFT_{1+1}$s relevant for the semiclassical large central charge analysis. There are two types of energy-momentum tensors in a $GCFT_{1+1}$ and the corresponding Galilean conformal Ward identities [67] look quite different from their relativistic counterparts. The finite $GCA_2$ transformations

$$t \to f(t), \quad x \to f'(t)x + g(t), \tag{74}$$

are generated by the Nöether charges [67]

$$M_n = \oint dt\, T_{tx}\, t^{n+1}, \quad L_n = \oint dt\, \left(T_{tt}\, t^{n+1} + (n+1)T_{tx}\, t^n x\right), \tag{75}$$

where $T_{\mu\nu}$ are the components of the $GCFT_{1+1}$ energy-momentum tensor. Inverting these relations, we obtain the components of the energy-momentum tensor as [67]

$$\mathcal{M} \equiv T_{tx} = \sum_n M_n\, t^{-n-2}, \quad \mathcal{L} \equiv T_{tt} = \sum_n \left[L_n + (n+2)\frac{x}{t}M_n\right] t^{-n-2}, \tag{76}$$

where $L_n$ and $M_n$ are the usual generators of GCA. Note that unlike the relativistic $CFT_2$s the two independent components of the energy-momentum tensor $\mathcal{L}$ and $\mathcal{M}$ have distinct functional forms in a $GCFT_{1+1}$. This is a reflection of the fact that the $GCA_2$, unlike the relativistic Virasoro algebra, does not decompose into two identical holomorphic and anti-holomorphic

---

[6]We have employed a shorthand notation for describing the coordinates $X_i = (x_i, t_i)$.

[7]This has to be contrasted with the $t$-channel $x \to 1, t \to 0$ for the $BMS_3$ field theory considered in [67]. We will use the methods developed in [67] to compute the Galilean conformal block utilizing the $BMS_3/GCA_2$ correspondence briefly discussed in section 3 which essentially demonstrates the equivalence of the two field theories under $x \leftrightarrow t$ [44].

copies. The Galilean conformal Ward identities obeyed by these two of energy-momentum tensors are given by [67,72]:

$$\langle \mathcal{M}(x,t)V_1(x_1,t_1)\dots V_n(x_n,t_n)\rangle = \sum_{i=1}^{n}\left[\frac{\chi_i}{(t-t_i)^2}+\frac{1}{t-t_i}\partial_{x_i}\right]\langle V_1(x_1,t_1)\dots V_n(x_n,t_n)\rangle,$$

$$\langle \mathcal{L}(x,t)V_1(x_1,t_1)\dots V_n(x_n,t_n)\rangle = \sum_{i=1}^{n}\left[\frac{\Delta_i}{(t-t_i)^2}-\frac{1}{t-t_i}\partial_{t_i}+\frac{2\chi_i(x-x_i)}{(t-t_i)^3}\right.$$
$$\left.+\frac{x-x_i}{(t-t_i)^2}\partial_{x_i}\right]\langle V_1(x_1,t_1)\dots V_n(x_n,t_n)\rangle,$$

(77)

where $V_i$ are GCFT$_{1+1}$ primaries, and $\chi_i$ and $\Delta_i$ are the corresponding scaling dimensions. We wish to analyze the large-$c_M$ limit of the following four-point function of twist operators in the $t$-channel described by $T \to 1$, $X \to 0$ [8]

$$\langle \Phi_{n_e}(X_1)\Phi_{-n_e}(X_2)\Phi_{-n_e}(X_3)\Phi_{n_e}(X_4)\rangle = \sum_{\alpha}\langle \Phi_{n_e}(X_1)\Phi_{n_e}(X_4)|\alpha\rangle\langle\alpha|\Phi_{-n_e}(X_2)\Phi_{-n_e}(X_3)\rangle \equiv \sum_{\alpha}\mathcal{F}_\alpha.$$

(78)

In eq. (78), $\mathcal{F}_\alpha$ are the GCA$_2$ conformal blocks corresponding to the $t$-channel and we have expanded the the four-point function into a basis of GCFT$_{1+1}$ primary operators denoted by the index $\alpha$. Figure 3 shows this expansion of the four-point function (78) in terms of Galilean partial waves. In the large central charge limit $c_M \to \infty$ the blocks $\mathcal{F}_\alpha$ are expected to have

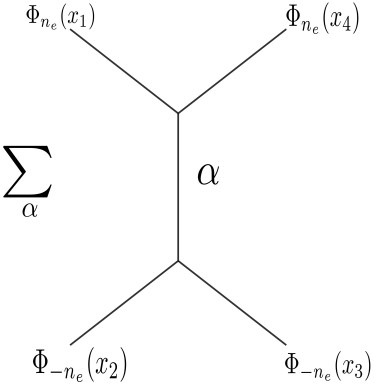

Figure 3: Galilean conformal block expansion of a four-point twist correlator in the $t$-channel. The choice of channel corresponds to two operators interchanging a GCA$_2$ highest weight representation with the other two. The exchanged representation is labeled by $\alpha$ which denotes primary operators in the theory.

an exponential structure similar to their relativistic counterparts [26,73]. In the following, we are going to perform a geometric monodromy analysis[9] in the semi-classical limit to obtain a large central charge expression for the Galilean conformal block $\mathcal{F}_\alpha$. Recall that unlike in the relativistic CFT$_{1+1}$s, the functional forms of the two energy-momentum tensor components in eq. (76) for a GCFT$_{1+1}$ are not identical and therefore we have to perform a separate monodromy analysis corresponding to each of them.

---

[8]$X$, $T$ are the usual cross ratios for the GCFT$_{1+1}$.

[9]Note that the monodromy analysis can also be formulated using the GCA$_2$ null vectors. The analysis will be a bit more involved than the relativistic case due to the presence of the so called GCA$_2$ multiplets [67]. Nevertheless the differential equations obtained via this technique will be the same as in the geometric monodromy method.

### 6.1.1  Monodromy of $\mathcal{M}$

In this subsection we will solve the differential equation for the expectation value of the energy-momentum tensor component $\mathcal{M}$. Subsequently we will utilize the monodromy technique developed in [67] to obtain a partial expression for the Galilean conformal block in eq. (78). Using the Ward identities in eq. (77) we obtain for the expectation value of the energy-momentum tensor $\mathcal{M}$ as

$$
\begin{aligned}
\mathcal{M}(X_i;X) &\equiv \frac{\left\langle \mathcal{M}(X)\Phi_{n_e}(X_1)\Phi_{-n_e}(X_2)\Phi_{-n_e}(X_3)\Phi_{n_e}(X_4)\right\rangle}{\left\langle \Phi_{n_e}(X_1)\Phi_{-n_e}(X_2)\Phi_{-n_e}(X_3)\Phi_{n_e}(X_4)\right\rangle} \\
&= \sum_{i=1}^{4}\left[\frac{\chi_i}{(t-t_i)^2} + \frac{c_M}{6}\frac{c_i}{t-t_i}\right],
\end{aligned}
\tag{79}
$$

where the auxiliary parameters are given by

$$
c_i = \frac{6}{c_M}\partial_{x_i}\log\left\langle \Phi_{n_e}(X_1)\Phi_{-n_e}(X_2)\Phi_{-n_e}(X_3)\Phi_{n_e}(X_4)\right\rangle.
\tag{80}
$$

The four-point function is not completely fixed by the conformal symmetry, and not all the auxiliary parameters $c_i$ are known. We will place the operators at $t_1 = 0$, $t_3 = 1$, $t_4 = \infty$ and leave $t_2 = T$ free. Requiring that the expectation value $\mathcal{M}(X_i;X)$ vanishes as $\mathcal{M}(T;t)\sim t^{-4}$ as $t\to\infty$ we obtain the conditions

$$
\sum_i c_i = 0,\quad \sum_i\left(\frac{c_M}{6}c_i t_i + \chi_i\right) = 0,\quad \sum_i\left(\frac{c_M}{6}c_i t_i^2 + 2\chi_i t_i\right) = 0.
\tag{81}
$$

Using the approximation that $\chi_i \equiv \chi_\Phi$, being the conformal dimension of the so called 'light' operator $\Phi_{n_e}$, vanishes when we take the replica limit $n_e \to 1$. This allows us to determine three of the auxiliary functions in terms of the remaining one as

$$
c_1 = c_2(T-1),\quad c_3 = -c_2 T,\quad c_4 = 0.
\tag{82}
$$

This leads to the following expression for the energy-momentum tensor expectation value

$$
\frac{6}{c_M}\mathcal{M}(T;t) = c_2\left[\frac{T-1}{t} + \frac{1}{t-T} - \frac{T}{t-1}\right].
\tag{83}
$$

The component $\mathcal{M}$ of the energy-momentum tensor transforms under a generic Galilean conformal transformation $x\to x'$, $t\to t'$ in eq. (74) as [67]

$$
\mathcal{M}'(t',x') = (f')^2\mathcal{M}(t,x) + \frac{c_M}{12}S(f,t),
\tag{84}
$$

where $S(f,t)$ is the Schwarzian derivative for the coordinate transformation $t\to f(t)$. Requiring the expectation value $\mathcal{M}(X_i;X)$ to vanish on the GCFT$_{1+1}$ plane for the ground state, this will lead to the condition

$$
\frac{1}{2}S(f,t) = c_2\left[\frac{T-1}{t} + \frac{1}{t-T} - \frac{T}{t-1}\right].
\tag{85}
$$

Eq. (85) is equivalent to the differential equation

$$
0 = h''(t) + \frac{1}{2}S(f,t)h(t) = h''(t) + \frac{6}{c_M}\mathcal{M}(T,t)h(t),
\tag{86}
$$

with $f = h_1/h_2$, $h_1$ and $h_2$ being the two solutions of the above differential equation. We will solve this equation by the method of variation of parameters up to linear order in the parameter $\epsilon_\alpha = \frac{6}{c_M}\chi_\alpha$. To zeroth order, setting $\mathcal{M}^{(0)} = 0$, the solutions are given by

$$h^{(0)}(t) = 1, t. \tag{87}$$

Therefore expanding up to linear order in $\epsilon_\alpha$

$$h_i = h_i^{(0)} + \epsilon_\alpha h_i^{(1)}, \quad \mathcal{M} = \mathcal{M}^{(0)} + \epsilon_\alpha \mathcal{M}^{(1)}, \tag{88}$$

the differential equation to solve up to this order is given by

$$h_i^{(1)''}(t) = -\frac{6}{c_M}\mathcal{M}^{(1)}(T,t) h_i^{(0)}(t). \tag{89}$$

After solving eq. (89) we compute the monodromy of the solutions by going around the light operators at $t = 1, T$ as described in [67] which leads to the following monodromy matrix:

$$M = \begin{pmatrix} 1 & 2\pi i\, c_2 T(T-1) \\ 2\pi i\, c_2(T-1) & 1 \end{pmatrix}. \tag{90}$$

Next we utilize the following monodromy condition for the three point twist correlator $\left\langle \Phi_{n_e}(x_1, t_1)\, \Phi_{n_e}(x_4, t_4)\, V_\alpha(X, T) \right\rangle$ obtained in appendix B,

$$\sqrt{\frac{I_1 - I_2}{2}} = 2\pi\epsilon_\alpha, \tag{91}$$

where $I_1 = \operatorname{tr} M$ and $I_2 = \operatorname{tr} M^2$ are invariant under global Galilean conformal transformations[10]. Using eq. (91) we can find the remaining auxiliary parameter $c_2$ as

$$c_2 = \epsilon_\alpha \frac{1}{\sqrt{T(T-1)}}. \tag{92}$$

Therefore the conformal block for the four-point function in eq. (78) may be obtained as:

$$\begin{aligned} \mathcal{F}_\alpha &= \exp\left[\frac{c_M}{6}\int c_2\, dX\right] \\ &= \exp\left[\chi_\alpha\left(\frac{X}{\sqrt{T(T-1)}}\right)\right]\tilde{\mathcal{F}}(T). \end{aligned} \tag{93}$$

Expression (93) for the Galilean conformal block still has an unknown function $\tilde{\mathcal{F}}(T)$. To determine $\tilde{\mathcal{F}}(T)$ we need to perform the monodromy analysis for the other energy-momentum tensor $\mathcal{L}$, which we will do in the next subsection. For the particular four-point function of twist correlators we consider in this section, we do not need to explore the monodromy for $\mathcal{L}$. The reason is that, since the conformal dimensions $\Delta_\Phi = \Delta_{n_e} \propto c_L$, they will vanish as long as we consider Einstein gravity for which eq. (21) gives $c_L = 0$. Therefore the monodromy problem for the energy-momentum tensor $\mathcal{L}$ becomes trivial and leads to $\tilde{\mathcal{F}}(T) = 1$. Nevertheless, in the next subsection we will explicitly solve the differential equation for $\mathcal{L}$ monodromy and show that this is indeed the case.

---

[10]Note that the condition in eq. (91) is valid in the leading order in the expansion parameter $\epsilon_\alpha$. For generic conformal dimensions $\chi_\alpha$ of the exchanged operator, the linear analysis may fail to capture the full monodromy of the solution and one needs to go beyond leading order. In appendix B, we have performed the next to leading order analysis and no further corrections to the conformal block in eq. (93) is found.

### 6.1.2 Monodromy of $\mathcal{L}$

To get the full expression of the Galilean conformal block, we will next focus on the monodromy problem for the energy-momentum tensor $\mathcal{L}$. We start with the expectation value of the energy-momentum tensor $\mathcal{L}$ inside the four-point correlator [67]

$$\mathcal{L}(X_i; X) \equiv \frac{\left\langle \mathcal{L}(X) \Phi_{n_e}(X_1) \Phi_{-n_e}(X_2) \Phi_{-n_e}(X_3) \Phi_{n_e}(X_4) \right\rangle}{\left\langle \Phi_{n_e}(X_1) \Phi_{-n_e}(X_2) \Phi_{-n_e}(X_3) \Phi_{n_e}(X_4) \right\rangle}. \tag{94}$$

Using the shorthands $\delta_i = \frac{c_M}{6} \Delta_i$ and $\epsilon_i = \frac{c_M}{6} \chi_i$, eq. (94) can be rewritten utilizing the Ward identities in eq. (77) as

$$\frac{6}{c_M} \mathcal{L}(X_i; (x, t)) = \sum_{i=1}^{4} \left[ \frac{\delta_i}{(t - t_i)^2} - \frac{1}{t - t_i} d_i + \frac{2\epsilon_i(x - x_i)}{(t - t_i)^3} + \frac{x - x_i}{(t - t_i)^2} c_i \right], \tag{95}$$

where the auxiliary parameters $c_i$ are defined in eq. (80) and $d_i$ admit similar definitions [67]:

$$d_i = \frac{6}{c_M} \partial_{t_i} \log \left\langle \Phi_{n_e}(X_1) \Phi_{-n_e}(X_2) \Phi_{-n_e}(X_3) \Phi_{n_e}(X_4) \right\rangle. \tag{96}$$

The smoothness of the expectation value $\mathcal{L}(X_i, X)$ requires $\mathcal{L}(T, t) \to t^{-4}$ as $t \to \infty$. Together with the freedom provided by global Galilean conformal transformations, this fixes all of the auxiliary parameters $d_i$ except one. Using the global Galilean conformal symmetry, we will place the operators at $t_1 = 0$, $t_2 = T$, $t_3 = 1$, $t_4 = \infty$ and $x_1 = 0$, $x_2 = X$, $x_3 = 0$ and $x_4 = 0$. This leads to the following values for three of the auxiliary parameters $d_i$ in terms of the remaining one:

$$\begin{aligned} d_1 &= c_2 X + d_2(T - 1) - 2\delta_L, \\ d_3 &= c_2(-X) - d_2 T + 2\delta_L, \\ d_4 &= 0, \end{aligned} \tag{97}$$

where $\delta_L = c_M \Delta_{n_e}/6$ and $\epsilon_L = c_M \chi_{n_e}/6$ denote the rescaled scaling dimensions of the twist operator $\Phi_{n_e}$. Substituting equations (97) and (80), into eq. (95) we obtain the expectation value $\mathcal{L}(X_i, (x, t))$ as

$$\begin{aligned} \frac{6}{c_M} \mathcal{L}(X_i; (x, t)) = &- \frac{c_2 X + d_2(T - 1) - 2\delta_L}{t} + \frac{c_2 X + d_2 T - 2\delta_L}{t - 1} + \frac{c_1 x}{t^2} \\ &+ \frac{c_2(x - X)}{(t - T)^2} + \frac{c_3 x}{(t - 1)^2} - \frac{d_2}{t - T} + \frac{2x\epsilon_L}{t^3} + \frac{\delta_L}{t^2} + \frac{\delta_L}{(t - 1)^2} + \frac{\delta_L}{(t - T)^2} \\ &+ \frac{2\epsilon_L(x - X)}{(t - T)^3} + \frac{2x\epsilon_L}{(t - 1)^3}. \end{aligned} \tag{98}$$

The transformation of the energy-momentum tensor $\mathcal{L}$ under the finite Galilean conformal transformation in eq. (2), leads to the following differential equation

$$\begin{aligned} \frac{6}{c_M} \mathcal{L}(X_i; (x, t)) = &\frac{g'\left(f'f'' - 3(f'')^3\right) + f'\left(3g''f'' - g'''f'\right)}{2(f')^3} \\ &- \frac{x\left(3(f'')^2 + f'''(f')^2 - 4f'''f'f''\right)}{2(f')^3}. \end{aligned} \tag{99}$$

As in [67], we now take the following combination of the expectation values

$$
\begin{aligned}
\frac{6}{c_M}\tilde{\mathcal{L}}(X_i;(x,t)) &= \frac{6}{c_M}\left[\mathcal{L}(X_i;(x,t)) + X\,\mathcal{M}'(X_i;(x,t))\right] \\
&= c_2 X\left(-\frac{1}{(t-T)^2} - \frac{1}{t} + \frac{1}{t-1}\right) - \frac{d_2(T-1)T}{(t-1)t(t-T)} \\
&\quad + \delta_L\left(\frac{1}{t^2} + \frac{1}{(t-T)^2} + \frac{2}{t} - \frac{2}{t-1} + \frac{1}{(t-1)^2}\right) + \frac{2X\epsilon_L}{(T-t)^3}\,.
\end{aligned}
\tag{100}
$$

Next we choose the ansatz $g(t) = f'(t)Y(t)$ for the coordinate transformation to reduce the differential equation in (99) to the following form:

$$
\frac{6}{c_M}\tilde{\mathcal{L}} = -\frac{1}{2}Y''' - 2Y'\frac{6}{c_M}\mathcal{M} - Y\frac{6}{c_M}\mathcal{M}'\,.
\tag{101}
$$

We can solve the above differential equation using the method described in [67] upto linear order of $\epsilon_\alpha$ and $\delta_\alpha$. The scaling dimensions of the light operator $\Phi_{n_e}$ vanishes when we take the replica limit $n_e \to 1$. After computing the monodromy by going around the light operators at $t = 1, T$, we obtain the auxiliary parameter $d_2$ as

$$
d_2 = \frac{(1-3T)X\epsilon_\alpha + 2(T-1)T\delta_\alpha}{2(T-1)^2 T^{3/2}}\,.
\tag{102}
$$

It is easy to check that the following is true from equations (80) and (96):

$$
\frac{\partial}{\partial X}d_2 = \frac{\partial}{\partial T}c_2\,.
\tag{103}
$$

Finally, we obtain the full Galilean conformal block using eq. (96) as

$$
\mathcal{F}_\alpha = \exp\left[\chi_\alpha\left(\frac{X}{\sqrt{T}(T-1)}\right)\right],
\tag{104}
$$

where we have used the fact that for $c_L = 0$, $\delta_\alpha$ vanishes. The complete Galilean conformal block in eq. (104) exactly matches with the $\mathcal{M}$ monodromy result in eq. (93) for $\tilde{\mathcal{F}}(T) = 1$ as anticipated before.

### 6.1.3 Entanglement negativity in the large-$c_M$ limit

In this subsection, we will use the large-$c_M$ limit of the $t$-channel Galilean conformal block in eq. (104) to compute the entanglement negativity for the bipartite mixed state of two disjoint intervals in proximity. Note from eq. (14) that, in the replica limit $n_e \to 1$ the scaling dimension of the twist field $\Phi_{n_e}$ vanishes rendering it to be a light operator in the large-$c_M$ limit. Following [41] we may write down the following operator product expansions in the GCFT$_{1+1}$

$$
\begin{aligned}
\Phi_{n_e}(x_1,t_1)\,\Phi_{-n_e}(x_2,t_2) &= \frac{k_{n_e}}{t_{12}^{2\Delta_{n_e}}}\exp\left[-2\chi_{n_e}\frac{x_{12}}{t_{12}}\right]\mathbb{1} + \dots, \quad (x_1,t_1)\to(x_2,t_2), \\
\Phi_{-n_e}(x_2,t_3)\,\Phi_{-n_e}(x_3,t_3) &= \frac{k_{n_e}}{t_{23}^{2\Delta_{n_e}}}\exp\left[-2\chi_{n_e}\frac{x_{23}}{t_{23}}\right]\Phi_{-n_e}^2 + \dots, \quad (x_2,t_2)\to(x_3,t_3).
\end{aligned}
\tag{105}
$$

Note from eq. (78) that in the $t$-channel described by $T \to 1, X \to 0$, the light operators which fuse together are located at $[(x_1,t_1),(x_4,t_4)]$ and $[(x_2,t_2),(x_3,t_3)]$, respectively. Therefore

utilizing eq. (105), it is easy to see that the dominant contribution to the four-point twist correlator in eq. (78) in the large-$c_M$ limit comes from the GCA$_2$ conformal block corresponding to the primary field $\Phi^2_{\pm n_e}$. Although it has the smallest conformal dimension, this twist operator remains heavy in the replica limit, $\chi_{n_e/2} \to -\frac{c_M}{16}$ (cf. footnote 4). Therefore, as in the usual relativistic CFT$_{1+1}$ setting described in [23,25], the partial wave expansion for the four-point twist correlator in eq. (78) is dominated by the exchange of $\Phi^2_{\pm n_e}$:

$$\mathcal{F}_{\chi^{(2)}_{n_e}} = \exp\left( -\frac{c_M}{8} \frac{X}{\sqrt{T}(T-1)} \right). \tag{106}$$

Finally, using equations (18), (19) and (78), we obtain the negativity in the large $c_M$-limit to be

$$\mathcal{E} = \log\left( \mathcal{F}_{\chi^{(2)}_{n_e}} \right) \approx \frac{c_M}{8} \frac{X}{1-T}, \tag{107}$$

where, we have used the fact that in $t-$channel $T \to 1$, and neglected the square-root in the denominator. Note that this expression is in terms of the cross ratio in the $t$-channel, $X/(1-T)$. In terms of the coordinates $(x_i, t_i)$ of the endpoints of the two disjoint intervals under consideration, the cross ratio is given by

$$\frac{X}{1-T} = \frac{x_{13}}{t_{13}} + \frac{x_{24}}{t_{24}} - \frac{x_{14}}{t_{14}} - \frac{x_{23}}{t_{23}}. \tag{108}$$

Therefore the entanglement negativity for two disjoint intervals $A_1 = [(x_1, t_1), (x_2, t_2)]$ and $A_2 = [(x_3, t_3), (x_4, t_4)]$ in proximity is given by

$$\mathcal{E} = \frac{c_M}{8} \left( \frac{x_{13}}{t_{13}} + \frac{x_{24}}{t_{24}} - \frac{x_{14}}{t_{14}} - \frac{x_{23}}{t_{23}} \right). \tag{109}$$

We may now utilize the Galilean conformal transformations from the GCFT$_{1+1}$ plane to the spatially compactified cylinder to obtain the entanglement negativity in the finite-sized system described by a GCFT$_{1+1}$ defined on a cylinder with circumference $L_\phi$. The result is

$$\mathcal{E} = \frac{c_M \pi}{8 L_\phi} \left[ u_{13} \cot\left( \frac{\pi \phi_{13}}{L_\phi} \right) + u_{24} \cot\left( \frac{\pi \phi_{24}}{L_\phi} \right) - u_{14} \cot\left( \frac{\pi \phi_{14}}{L_\phi} \right) - u_{23} \cot\left( \frac{\pi \phi_{23}}{L_\phi} \right) \right]. \tag{110}$$

Finally we compute the entanglement negativity for the two disjoint intervals in a thermal GCFT$_{1+1}$ living on a cylinder of circumference $\beta$, where $\beta$ is the inverse temperature. We obtain the following expression for the entanglement negativity

$$\mathcal{E} = \frac{c_M \pi}{8 \beta} \left[ u_{13} \coth\left( \frac{\pi \phi_{13}}{\beta} \right) + u_{24} \coth\left( \frac{\pi \phi_{24}}{\beta} \right) - u_{14} \coth\left( \frac{\pi \phi_{14}}{\beta} \right) - u_{23} \coth\left( \frac{\pi \phi_{23}}{\beta} \right) \right]. \tag{111}$$

We will use these expressions for the entanglement negativity of two disjoint intervals in proximity to propose a holographic conjecture to obtain the same from the bulk computations.

## 6.2 Holographic entanglement negativity for two disjoint intervals in proximity

In this subsection we will advance a holographic proposal for computing the entanglement negativity of the bipartite mixed state configuration of two disjoint intervals in proximity in a holographic GCFT$_{1+1}$. According to the flat space holography, the GCFT$_{1+1}$ is dual to a bulk asymptotically flat spacetime. As before, we consider two disjoint Galilean boosted intervals $A_1 = [(x_1, t_1), (x_2, t_2)]$ and $A_2 = [(x_3, t_3), (x_4, t_4)]$ in the ground state of a holographic GCFT$_{1+1}$. The subsystem $A = A_1 \cup A_2$ is in a mixed state, and the separation between $A_1$ and

$A_2$, denoted $A_s$, belongs to the complementary subsystem $B = A^c$. As the flat holographic proposals in equations (55) and (67) for a single and two disjoint intervals turned out to have exactly the same functional form as their relativistic counterparts in [20, 21], we expect a similar holographic connection for the present configuration as well.

We will make use of the monodromy computations in the previous subsection 6.1.3 to justify our proposal. To this end we start with the following expression for the two point twist correlator in a holographic GCFT$_{1+1}$ on the plane (cf. eq. (7)):

$$\left\langle \Phi_{n_e}(x_1, t_1)\Phi_{-n_e}(x_2, t_2)\right\rangle \sim \exp\left(-2\chi_{n_e}\frac{x_{12}}{t_{12}}\right), \tag{112}$$

where we have used eq. (21) and eq. (14) to set $\Delta_{n_e} = 0$. Now we utilize the holographic dictionary in eqs. (24) and (25), to write eq. (106) as

$$\begin{aligned}\left\langle \Phi_{n_e}(x_1, t_1)\Phi_{-n_e}(x_2, t_2)\Phi_{-n_e}(x_3, t_3)\Phi_{n_e}(x_4, t_4)\right\rangle &\simeq \exp\left[\frac{c_M}{8}\left(\frac{x_{13}}{t_{13}} + \frac{x_{24}}{t_{24}} - \frac{x_{14}}{t_{14}} - \frac{x_{23}}{t_{23}}\right)\right]\\ &= \exp\left[\frac{c_M}{16}\left(L_{13}^{\text{extr}} + L_{24}^{\text{extr}} - L_{14}^{\text{extr}} - L_{23}^{\text{extr}}\right)\right],\end{aligned} \tag{113}$$

where in the second equality we have made use of eq. (45). We now propose, based on the monodromy computations in section 6.1.3, the following conjecture for the holographic entanglement negativity of two disjoint intervals in proximity located at the null infinity of the bulk asymptotically flat spacetime dual to a GCFT$_{1+1}$:

$$\begin{aligned}\mathcal{E} &= \frac{3}{16G}\left(L_{13}^{\text{extr}} + L_{24}^{\text{extr}} - L_{14}^{\text{extr}} - L_{23}^{\text{extr}}\right)\\ &= \frac{3}{16G}\left(L_{A_1 \cup A_s}^{\text{extr}} + L_{A_s \cup A_2}^{\text{extr}} - L_{A_1 \cup A_2 \cup A_s}^{\text{extr}} - L_{A_s}^{\text{extr}}\right),\end{aligned} \tag{114}$$

where $c_L = 0$ and $c_M = \frac{3}{G}$. Once again we observe that the holographic entanglement negativity for the mixed state configuration of two disjoint intervals in a holographic GCFT$_{1+1}$ involves a specific linear combination of the lengths of bulk extremal curves homologous to the intervals as shown in figure 4. Remarkably our flat holographic conjecture in eq. (114) has exactly the same structure as its relativistic counterpart in the AdS$_3$/CFT$_2$ scenario obtained in [23, 29]. It is interesting to note that, in the limit of adjacent intervals $x_{23} \to \epsilon$, where $\epsilon$ is the UV cut-off ($L_{A_s}^{\text{extr}} \to 0$ in the bulk), we get back our formula for two adjacent intervals in eq. (67). This serves as a strong consistency check of our proposal. Now we make use of the flat version of the HRT formula in eq. (26) to recast our formula for holographic entanglement negativity in the following instructive form

$$\begin{aligned}\mathcal{E} &= \frac{3}{4}\left(S_{A_1 \cup A_s} + S_{A_s \cup A_2} - S_{A_1 \cup A_2 \cup A_s} - S_{A_s}\right)\\ &= \frac{3}{4}\left(\mathcal{I}(A_1 \cup A_s; A_2) + \mathcal{I}(A_s; A_2)\right).\end{aligned} \tag{115}$$

Therefore we see that our holographic conjecture relates two very different entanglement measures, namely, entanglement negativity which is the upper bound of distillable entanglement, and the mutual information which measures entanglement correlation between two subsystems. Again, this particular connection seems unique for the specific configuration of two disjoint intervals on the boundary field theory. Interestingly, in the limit of adjacent interval $A_s \to \emptyset$ we get back the adjacent formula in eq. (68).

In the following, we are going to employ our holographic conjecture to compute the entanglement negativities in various configurations described by two disjoint intervals in proximity in different mixed states of a holographic GCFT$_{1+1}$. Remarkably our formula reproduces the universal behaviour of the holographic entanglement negativity at the large central charge limit of the holographic GCFT$_{1+1}$.

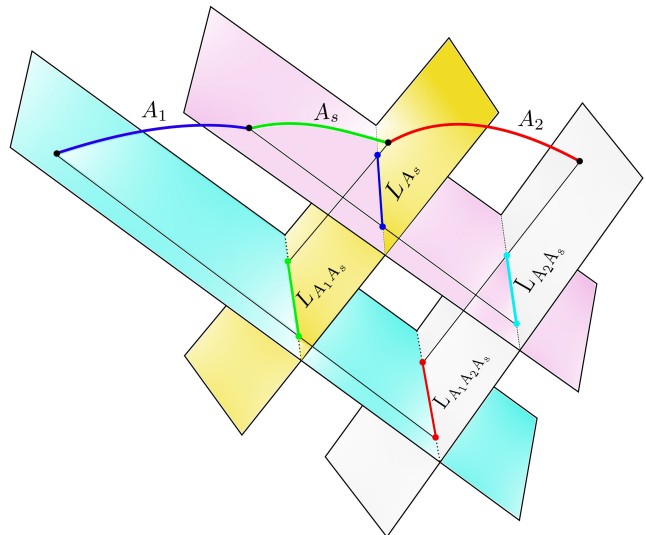

Figure 4: Schematics of the holographic construction for the computation of entanglement negativity of two disjoint intervals. The entanglement negativity is obtained via a specific linear combination of the lengths of the bulk extremal curves situated at the crossings of the null planes descending from the endpoints of the two intervals.

### 6.2.1 Two disjoint intervals in vacuum

We start with the mixed state configuration of two disjoint intervals $A_1 = [(x_1, t_1), (x_2, t_2)]$ and $A_2 = [(x_3, t_3), (x_4, t_4)]$ in the ground state of a holographic GCFT$_{1+1}$. The dual bulk geometry is that of pure Minkowski spacetime. Utilizing eq. (45) for the length of the extremal geodesics in locally Minkowski geometry, one obtain for the holographic entanglement negativity from eq. (114) as

$$
\begin{aligned}
\mathcal{E} &= \frac{3}{8G} \left( \frac{x_{13}}{t_{13}} + \frac{x_{24}}{t_{24}} - \frac{x_{14}}{t_{14}} - \frac{x_{23}}{t_{23}} \right) \\
&= \frac{c_M}{8} \left( \frac{l_1 + l_s}{t_1 + t_s} + \frac{l_2 + l_s}{t_2 + t_s} - \frac{l_1 + l_2 + l_s}{t_1 + t_2 + t_s} - \frac{l_s}{t_s} \right),
\end{aligned}
\tag{116}
$$

where we have denoted $l_1 = x_1 - x_2$, $l_s = x_2 - x_3$ and $l_2 = x_3 - x_4$ for the lengths of the respective intervals (cf. figure 4) and similarly for $t_1$, $t_2$ and $t_s$. remarkably this matches exactly with the large central charge behaviour of the entanglement negativity in eq. (109) obtained using the monodromy method in subsection 6.1.3. Considering the adjacent limit $l_s \to \epsilon$ and $t_s \to \epsilon$ (where $\epsilon$ is the UV cut-off) and taking the leading order terms in $\epsilon$, we get back the result for entanglement negativity for adjacent intervals in eq. (69).

### 6.2.2 Two disjoint intervals at a finite temperature

Next we will consider the mixed state configuration of two disjoint intervals in a thermal GCFT$_{1+1}$ living on a cylinder compactified in the timelike direction with circumference $\beta$. The dual spacetime is the locally FSC geometry described in subsection 5.1.3. Substituting eq. (57) for the length of the extremal curve in FSC geometry in our holographic conjecture in eq. (114) we obtain for the holographic entanglement negativity of two disjoint intervals at a

finite temperature

$$
\begin{aligned}
\mathcal{E} &= \frac{3\pi}{8G\beta}\left(u_{13}\coth\left(\frac{\pi\phi_{13}}{\beta}\right) + u_{24}\coth\left(\frac{\pi\phi_{24}}{\beta}\right) - u_{14}\coth\left(\frac{\pi\phi_{14}}{\beta}\right) - u_{23}\coth\left(\frac{\pi\phi_{23}}{\beta}\right)\right) \\
&= \frac{c_M}{8}\frac{\pi}{\beta}\left[(t_1+t_s)\coth\left(\frac{\pi(l_1+l_s)}{\beta}\right) + (t_2+t_s)\coth\left(\frac{\pi(l_2+l_s)}{\beta}\right)\right. \\
&\quad \left. - (t_1+t_2+t_s)\coth\left(\frac{\pi(l_1+l_2+l_s)}{\beta}\right) - t_s\coth\left(\frac{\pi l_s}{\beta}\right)\right],
\end{aligned}
$$

(117)

where the lengths of the respective intervals are denoted by $l_1 = u_1 - u_2$, $l_s = u_2 - u_3$ and $l_2 = u_3 - u_4$, and the times are given by $t_1$, $t_2$ and $t_s$. Again this matches exactly with the field theory computations at large central charge limit in eq. (110). We may take the adjacent limit $l_s \to \epsilon$ and $t_s \to \epsilon$, to show that the leading order expression matches exactly with the result for two adjacent intervals given in eq. (70).

### 6.2.3 Two disjoint intervals in a finite-sized system

Finally we turn our attention to the holographic computation of the entanglement negativity for two disjoint intervals in a finite-sized system obeying periodic boundary conditions described by a GCFT$_{1+1}$ living on a cylinder of circumference $L_\phi$ compactified along the spatial direction. The bulk dual is again asymptotically flat and is described by the global Minkowski orbifold metric in eq. (28). We now employ the expression for the extremal geodesic length in such spacetimes from eq. (36) to obtain the following expression for the entanglement negativity of the mixed state configuration described by two disjoint intervals in a finite-sized system as

$$
\begin{aligned}
\mathcal{E} &= \frac{3\pi}{8GL_\phi}\left(u_{13}\cot\left(\frac{\pi\phi_{13}}{L_\phi}\right) + u_{24}\cot\left(\frac{\pi\phi_{24}}{L_\phi}\right) - u_{14}\cot\left(\frac{\pi\phi_{14}}{L_\phi}\right) - u_{23}\cot\left(\frac{\pi\phi_{23}}{L_\phi}\right)\right) \\
&= \frac{c_M}{8}\frac{\pi}{L_\phi}\left[(t_1+t_s)\cot\left(\frac{\pi(l_1+l_s)}{L_\phi}\right) + (t_2+t_s)\cot\left(\frac{\pi(l_2+l_s)}{L_\phi}\right)\right. \\
&\quad \left. - (t_1+t_2+t_s)\cot\left(\frac{\pi(l_1+l_2+l_s)}{L_\phi}\right) - t_s\cot\left(\frac{\pi l_s}{L_\phi}\right)\right].
\end{aligned}
$$

(118)

Remarkably this again matches exactly with the field theory result in eq. (111) obtained through large central charge computations in subsection 6.1.3. Again in the adjacent limit described by $l_s \to \epsilon$ and $t_s \to \epsilon$, we get back the adjacent intervals result in eq. (71).

## 7 Holographic entanglement negativity in flat space TMG

In the previous sections we have computed the holographic entanglement negativity in the case of Einstein gravity in the bulk for which the dual GCFT$_{1+1}$ at the boundary had only one non-vanishing central charge $c_M$. At this point, we recall the fact that the representations of the GCA$_2$ algebra are labelled by the quantum numbers $\Delta$ and $\chi$. Therefore a vanishing $c_L$ would correspond to $\Delta = 0$ which describes a spinless massive particle propagating in the asymptotically flat bulk spacetime.

In this section we will incorporate the effects of a non-zero $c_L$, and hence a non-zero $\Delta$, in the bulk in order to see the agreement with the field theory results in [55] more closely. We expect that a non-vanishing $\Delta$ would introduce a spin for the massive particle. In this context we modify the bulk picture by introducing Topologically Massive Gravity (TMG) [59,

60, 63–66] which contains a gravitational Chern-Simons (CS) term. This Chern-Simons term arises due to a gravitational anomaly present in the relativistic $CFT_2$ whose İnönü-Wigner contraction leads to the $GCFT_{1+1}$s considered in the present article. From the perspective of the bulk, the dual operation to this parametric contraction on the boundary corresponds to taking the flat limit of the bulk $AdS_3$ geometry. Therefore the flat-holographic connection between TMG in asymptotically flat spacetimes and $GCFT_{1+1}$s with non-vanishing $c_L$ and $c_M$ comes from two equivalent parametric contractions of each sector in the original TMG-$AdS_3$/$CFT_2$ correspondence [59, 60, 64–66].

We start by briefly reviewing the salient features of TMG in $AdS_3$ spacetimes. The action of TMG in $AdS_3$ is the sum of the usual Einstein-Hilbert term, the cosmological constant term and a gravitational Chern-Simons term [59, 66] [11]:

$$
\begin{aligned}
\mathcal{S}_{\text{TMG}} &= \mathcal{S}_{\text{EH}} + \frac{1}{\mu} \mathcal{S}_{\text{CS}} \\
&= \frac{1}{16\pi G} \int d^3x \ \sqrt{-g} \left[ R + \frac{2}{\ell^2} + \frac{1}{2\mu} \varepsilon^{\alpha\beta\gamma} \left( \Gamma^{\rho}_{\alpha\sigma} \partial_\beta \Gamma^{\sigma}_{\gamma\rho} + \frac{2}{3} \Gamma^{\rho}_{\alpha\sigma} \Gamma^{\sigma}_{\beta\eta} \Gamma^{\eta}_{\gamma\rho} \right) \right],
\end{aligned}
\tag{119}
$$

where $\mu$ has mass dimension one and describes the coupling of the CS-term, and $\ell$ is the $AdS_3$ radius. In the limit $\mu \to \infty$ one recovers Einstein gravity. The asymptotic symmetry analysis of TMG in $AdS_3$ shows that the algebra of the modes of the asymptotic Killing vectors is isomorphic to two copies of Virasoro algebra with left and right moving central charges [59, 66]:

$$
c^{+}_{\text{TMG}} = \frac{3\ell}{2G} \left( 1 + \frac{1}{\mu\ell} \right), \quad c^{-}_{\text{TMG}} = \frac{3\ell}{2G} \left( 1 - \frac{1}{\mu\ell} \right).
\tag{120}
$$

Now we will go to asymptotically flat spacetime by taking the flat limit $\ell \to \infty$ leading to the flat space TMG. Remarkably the asymptotic symmetry group analysis at null infinity leads to the Galilean conformal algebra, with both central charges non-vanishing [45, 48, 59, 60]:

$$
c_L = \frac{3}{\mu G}, \qquad c_M = \frac{3}{G}.
\tag{121}
$$

Alternatively, these central charges can be obtained from $AdS_3$ by taking İnönü-Wigner contraction [59]: $c_L = c^{+}_{\text{TMG}} - c^{-}_{\text{TMG}}, c_M = (c^{+}_{\text{TMG}} + c^{-}_{\text{TMG}})/\ell$. From eq. (121) it is easy to see that in the limit $\mu \to \infty$ we get back Einstein gravity in asymptotically flat spacetime.

## 7.1 Extrapolating the holographic dictionary

In [66], the authors computed the holographic entanglement entropy for a $CFT_2$ with gravitational anomaly using the theory of topologically massive gravity in $AdS_3$. It was found that the difference in the left and right moving central charges of the anomalous $CFT_2$ gives rise to a non-trivial spin of the twist operators in the replica manifold, which in the context of $AdS_3$/$CFT_2$, corresponds to a massive spinning particle of mass $m = \chi$ and spin $s = \Delta$ moving in the bulk geometry of TMG-$AdS_3$. As easily seen from the action in eq. (119), the Chern-Simons term is unaffected by the flat limit $\ell \to \infty$ and therefore the above discussion remains valid in the flat-holographic scenario as well [60]. The action of such a particle was found to be [60, 66]:

$$
S_{\text{flat-TMG}} = \int_C ds \left( \chi \sqrt{\eta_{\mu\nu} \dot{X}^\mu \dot{X}^\nu} + \Delta (\tilde{n}.\nabla n) \right) + S_{\text{constraints}},
\tag{122}
$$

where $\tilde{n}$ and $n$ are unit space-like and time-like vectors respectively, both normal at the trajectory of the particle $X^\mu$, and $S_{\text{constraints}}$ is an action imposing these constraints through Langrange multipliers [60, 66]. In eq. (122) $C$ denotes the worldline of the particle. The action

---

[11]This should be contrasted with the Chern-Simons gauge theory of 3d gravity put forward by Witten [61].

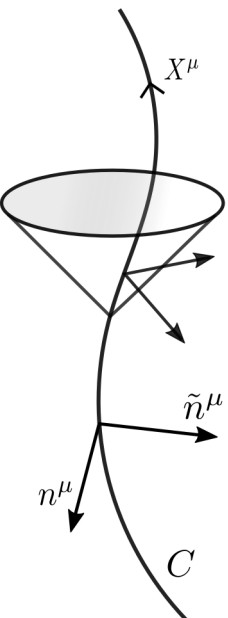

Figure 5: The topological Chern-Simons term in the TMG action introduces a normal frame defined by two auxiliary normal vectors $n$ and $\tilde{n}$ at each point on the worldline of a massive spinning particle. Figure modified from [66].

(122) introduces two new vectors in the the 3-dimensional bulk, while the constraint action $S_{\text{constraints}}$ imposes five constraints, leading to a single new degree of freedom. This sets up a normal frame to each point in the bulk as shown in fig. 5, and particle worldlines get broadened in the shape of ribbons [66]. The equations of motion reveal that this is not a true degree of freedom in the sense that the variations of the new vectors $n$ and $\tilde{n}$ along the worldline $X^\mu$ does not affect the action (122) [60, 66]. It is also interesting to note that straight lines governed by $\ddot{X}^\mu = 0$ in locally Minkowski spacetimes are still solutions of the equations of motion in the TMG background [60]. It is important to note that our holographic constructions for computing the entanglement negativity in terms of bulk geodesics rely heavily on the straight-line nature of the geodesics. To proceed, we note that in order to compute the entanglement entropy from the bulk perspective in a AdS/CFT setting, one considers the notion of the generalized gravitational entropy [17]. The computation of generalized gravitational entropy involves a replication of the dual gravitational geometry in the replica index $n$ followed by a quotienting through the replica symmetry $\mathbf{Z}_n$. In the quotient spacetime of the replicated geometry, there are conical defects along the entangling surfaces, namely at the endpoints of the boundary interval. We now propose, following [60, 66] that the two-point function of the twist fields inserted at the endpoints of the interval on the boundary of the quotient geometry is given by the exponential of the on-shell action of a massive spinning particle with mass $m_n = \chi_n$ and spin $s_n = \Delta_n$. For such a particle propagating along an extremal worldline in the bulk geometry from a point $x_i$ with a normal vector $n_i$ to a point $x_f$ with normal vector $n_f$, the two-point twist correlator has the form:

$$\left\langle \Phi_{n_e}(\partial_1 A) \Phi_{-n_e}(\partial_2 A) \right\rangle = e^{-\chi_{n_e} S_{\text{on-shell}}^{\text{EH}} - \Delta_{n_e} S_{\text{on-shell}}^{\text{CS}}}, \tag{123}$$

where

$$S_{\text{on-shell}}^{\text{EH}} = \sqrt{\eta_{\mu\nu} \dot{X}^\mu \dot{X}^\nu} = L^{\text{extr}}(x_i, x_f), \tag{124}$$

and $S^{\text{CS}}_{\text{on-shell}}$ is the topological Chern-Simons contribution to the on-shell action. As described before, the effect of this topological action is to broaden the worldline in the shape of a ribbon as the vectors $n$ and $\tilde{n}$ in eq. (122) define a normal frame to the curve $\mathcal{C}$. In eq. (123) the Chern-Simons contribution to the on-shell action in eq. (122) is given by the twist in the ribbon-shaped worldline as the particle moves along it [59, 60, 66]:

$$S^{\text{CS}}_{\text{on-shell}} = \int_{\mathcal{C}} ds\,(\tilde{n}.\nabla n) = \cosh^{-1}(-n_i.n_f). \tag{125}$$

Equation (125) essentially computes the boost $\Delta\eta$ required to drag the orthonormal frame generated by the vectors $(\dot{X}, n_i, n_f)$ from the point $x_i$ to $x_f$.

In the following subsection we will perform the computations of the spinning two-point correlators for different bulk geometries in flat space-TMG using the modified holographic dictionary in eqs. (123) to (125). With this generalized expression for the two point twist-correlator in eq. (123) all our previous analysis in section 5 will simply follow and lead to modified formulae for the holographic entanglement negativity in GCFT$_{1+1}$ dual to bulk geometries governed by TMG [12].

## 7.2 Two-point correlator of twist fields with spin

We start with TMG in a pure Minkowski spacetime. A schematics of the bulk geometry corresponding a single interval $A = [(x_1, t_1), (x_2, t_2)]$ in the boundary GCFT$_{1+1}$ is shown in fig. 6. We have two bulk normal vectors $n_i^{\partial}$ erected at each of the bulk points $y_i$ $(i = 1, 2)$ descending from the endpoints $(u_i, \phi_i)$[13] of the interval on the boundary, which were chosen in [60] to be pointed along the directions of the corresponding null rays $\gamma_i$:

$$\dot{\gamma}_1 = \partial_r\Big|_{\gamma_1} = \partial_t + \cos\phi_1\,\partial_x + \sin\phi_1\,\partial_y\,, \quad \dot{\gamma}_2 = \partial_r\Big|_{\gamma_2} = \partial_t + \cos\phi_2\,\partial_x + \sin\phi_2\,\partial_y\,. \tag{126}$$

Since these two vectors are null, the authors in [60] introduced two timelike vectors:

$$n_1 = \frac{1}{\epsilon}\dot{\gamma}_1 - \frac{\epsilon}{2}\frac{1}{\dot{\gamma}_1.\dot{\gamma}_2}\dot{\gamma}_2\,, \quad n_2 = \frac{1}{\epsilon}\dot{\gamma}_2 - \frac{\epsilon}{2}\frac{1}{\dot{\gamma}_1.\dot{\gamma}_2}\dot{\gamma}_1\,. \tag{127}$$

With these definitions we obtain from eq. (125) in the $\epsilon \to 0$ limit

$$S^{\text{CS}}_{\text{on-shell}} = \Delta\eta_{12} = \cosh^{-1}(-\frac{\dot{\gamma}_1.\dot{\gamma}_2}{\epsilon^2}) = \left|\log\left(-\frac{2\dot{\gamma}_1.\dot{\gamma}_2}{\epsilon^2}\right)\right| = 2\log\left(\frac{2}{\epsilon}\sin\frac{\phi_{12}}{2}\right). \tag{128}$$

In eq. (128) the boost $\Delta\eta_{12}$ may be interpreted as the difference in the twist of the two endpoints of the ribbon-like geometries induced by the topological term in eq. (125). Therefore the two-point spinning twist correlator in eq. (123) in the case of pure Minkowski spacetime dual to a GCFT$_{1+1}$ in its ground state is given by

$$\left\langle \Phi_{n_e}(\partial_1 A)\Phi_{-n_e}(\partial_2 A)\right\rangle = \left(\frac{2}{\epsilon}\sin\frac{\phi_{12}}{2}\right)^{-2\Delta_{n_e}}\exp\left(-\chi_{n_e}\frac{u_{12}}{\tan\frac{\phi_{12}}{2}}\right), \tag{129}$$

where we have used eq. (23) for the extremal geodesic length and $\partial_i A = (u_i, \phi_i)$ denotes the entangling surfaces, namely, the endpoints of the interval at the boundary.

---

[12]All these results may be recast in the factorised Wilson line prescription in the Chern-Simons formulation of 3d gravity developed in [66].

[13]$(u_i, \phi_i)$ are the cylindrical coordinates related to the planar coordinates $(x_i, t_i)$ via eq. (5).

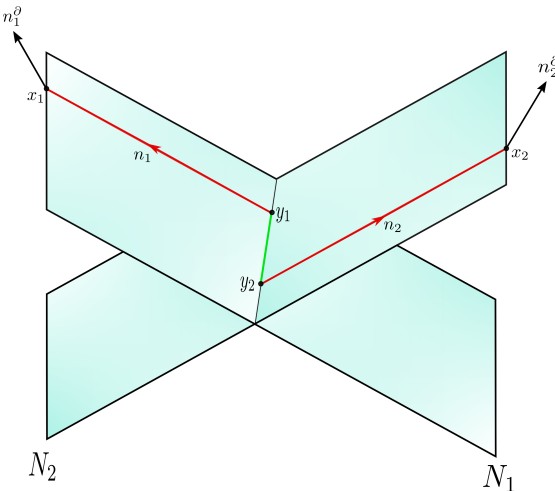

Figure 6: Bulk setup for computing two-point correlator of twist fields with non-zero spin. There are boundary normal vectors $n_i^\partial$ on each of the black points on the asymptotic boundary. The black points are on the null curves descending from these boundary points and they are equipped with normal vectors $n_i \propto \partial_r\big|_{\gamma_i}$. Figure modified from [60].

Next we proceed to compute the boost in the case of non-rotating FSC geometry. In that case the bulk null vectors in eq. (126) become (cf. eq. (29))

$$
\begin{aligned}
\dot\gamma_1 &= \frac{\beta}{2\pi}\cosh\left(\frac{2\pi\phi_1}{\beta}\right)\partial_t + \frac{\beta}{2\pi}\sinh\left(\frac{2\pi\phi_1}{\beta}\right)\partial_x - \frac{\beta}{2\pi}\partial_y\,,\\
\dot\gamma_2 &= \frac{\beta}{2\pi}\cosh\left(\frac{2\pi\phi_2}{\beta}\right)\partial_t + \frac{\beta}{2\pi}\sinh\left(\frac{2\pi\phi_2}{\beta}\right)\partial_x - \frac{\beta}{2\pi}\partial_y\,.
\end{aligned}
\tag{130}
$$

Therefore using eqs. (125) and (128) we obtain

$$
\Delta\eta_{12}^{\mathrm{FSC}} = 2\log\left(\frac{\beta}{\pi\epsilon}\sinh\frac{\pi\phi_{12}}{\beta}\right).
\tag{131}
$$

Similar computations in the case of TMG in global Minkowski orbifold geometries yields

$$
\Delta\eta_{12}^{\mathrm{GM}} = 2\log\left(\frac{L_\phi}{\pi\epsilon}\sin\frac{\pi\phi_{12}}{L_\phi}\right).
\tag{132}
$$

In the following subsections we will utilize equations (128), (131) and (132) for the twists in the ribbon to compute the topological CS contribution to the holographic entanglement negativity for different sub-interval geometries in a holographic GCFT$_{1+1}$.

## 7.3 Holographic entanglement negativity for a single interval

In this subsection we will generalize the proposals (44) and (55) for computing entanglement negativity of various bipartite pure and mixed state configurations described by a single interval in a GCFT$_{1+1}$ to incorporate the non-vanishing $c_L$ effects. To this end, we first consider the pure state configurations described by a single interval $A = [(x_1, t_1), (x_2, t_2)]$ in the ground state of a GCFT$_{1+1}$ at zero temperature. To proceed, we replace eq. (44) for the two-point function of twist operators by the corresponding expression with non-zero spin in eq. (129).

Now, using the modified holographic dictionary in equations (123), (124) and (125) we write the two-point function of the composite twist operators $\Phi^2_{n_e}$ inserted at the endpoints of the single interval $A = [(x_1, t_1), (x_2, t_2)]$ as:

$$\left\langle \Phi^2_{n_e}(x_1, t_1) \Phi^2_{-n_e}(x_2, t_2) \right\rangle = \exp\left[-2\,\chi_{n_e/2}\, L^{\mathrm{extr}}_{12} - 2\Delta_{n_e/2}\,\Delta\eta_{12}\right], \tag{133}$$

where $L^{\mathrm{extr}}_{12}$ is the length of the extremal ribbon-shaped curve anchored on the entangling surfaces, and $\Delta\eta_{12}$ denotes the difference in the twist at the endpoints of the ribbon. Now the entanglement negativity for the pure state configuration described by the single interval in the GCFT$_{1+1}$ vacuum may be obtained from eq. (42) as

$$\mathcal{E} = \frac{3}{8G}\left(\mathrm{L}^{\mathrm{extr}}_{12} + \frac{1}{\mu}\Delta\eta_{12}\right), \tag{134}$$

where we have used equations (14) and (121) and subsequently took the replica limit. In the following, we will make use of the holographic formula in eq. (134) to compute the holographic entanglement negativity for the bipartite pure state configurations described by a single interval in the vacuum state of a holographic GCFT$_{1+1}$ as well as for a GCFT$_{1+1}$ describing a system of finite size. Later, we will consider the mixed state configuration of a single interval at a finite temperature which involves an analysis of a particular four-point twist correlator in the large central charge limit in the spirit of subsection 5.1.3.

### 7.3.1 Single interval at zero temperature

We start with the simplest pure state configuration of a single interval in the vacuum state of a holographic GCFT$_{1+1}$ at zero temperature for which the dual bulk geometry corresponds to the pure Minkowski spacetime. Utilizing the transformations (5), the CS-contribution to the two-point function in eq. (128) may be written in the planner coordinates as:

$$\Delta\eta_{12} = 2\log\left(\frac{t_{12}}{\epsilon}\right). \tag{135}$$

We now substitute equations (23) and (135) in eq. (134) to obtain the holographic entanglement negativity as

$$\mathcal{E} = \frac{c_L}{4}\log\left(\frac{t_{12}}{\epsilon}\right) + \frac{c_M}{4}\frac{x_{12}}{t_{12}}, \tag{136}$$

where we have used eq. (121) for the central charges of the holographic GCFT$_{1+1}$. Remarkably, we have reproduced the universal part of the complete result obtained in [55] via replica technique.

### 7.3.2 Single interval in a finite-sized system

Next we move on to the computation of the holographic entanglement negativity for the bipartite pure state configuration of a single interval in a GCFT$_{1+1}$ describing a finite-sized system endowed with periodic boundary conditions. The corresponding bulk geometry is described by the global Minkowski orbifold with metric (28). Using the expression for the corresponding length of the extremal curve in eq. (36), and the twist in eq. (132), we obain the holographic entanglement entropy from eq. (134) as

$$\mathcal{E} = \frac{c_L}{4}\log\left[\frac{L_\phi}{\pi\epsilon}\sin\left(\frac{\pi\phi_{12}}{L_\phi}\right)\right] + \frac{c_M}{4}\frac{\pi u_{12}}{L_\phi}\cot\left(\frac{\pi\phi_{12}}{L_\phi}\right). \tag{137}$$

This matches exactly with the universal part of the complete field theory result in [55].

### 7.3.3 Single interval at a finite temperature

Finally we focus on the mixed state configuration of a single interval at a finite temperature. The field theory is described by a thermal $GCFT_{1+1}$ on a cylinder compactified along the timelike direction with circumference $\beta$. As described in subsection 5.1.3, the definition of the holographic entanglement negativity for this configuration involves two auxiliary intervals $B_1$ and $B_2$ sandwiching the single interval $A$. This leads to a four-point twist correlator which admits a large central charge factorization of the form (52). For a thermal $GCFT_{1+1}$ with unequal non-vanishing central charges (120), the dual gravitational theory is described by topologically massive gravity in FSC geometries. For such $GCFT_{1+1}$s, using the modified flat-holographic dictionary in eq. (123), the four-point twist correlator in eq. (52) has the large-central charge structure:

$$
\begin{aligned}
&\left\langle \Phi_{n_e}(x_1,t_1)\Phi^2_{-n_e}(x_2,t_2)\Phi^2_{n_e}(x_3,t_3)\Phi_{-n_e}(x_4,t_4)\right\rangle \\
&= \exp\Big[-\chi_{n_e}\mathrm{L}^{extr}_{14}-\chi_{n_e/2}\Big(2\mathrm{L}^{extr}_{23}+\mathrm{L}^{extr}_{12}+\mathrm{L}^{extr}_{34}-\mathrm{L}^{extr}_{13}-\mathrm{L}^{extr}_{24}\Big) \\
&\qquad -\Delta_{n_e}\Delta\eta_{14}-\Delta_{n_e/2}\Big(2\Delta\eta_{23}+\Delta\eta_{12}+\Delta\eta_{34}-\Delta\eta_{13}-\Delta\eta_{24}\Big)\Big],
\end{aligned}
\tag{138}
$$

where $\mathrm{L}^{extr}_{ij}$ are the lengths of the extremal ribbon-shaped curves anchored on various subsystems constituted by the single interval $A$ and the auxiliary intervals $B_1, B_2$, and $\eta_{ij}$ are the corresponding twists in the ribbons. Taking the replica limit $n_e \to 1$ followed by the bipartite limit $B_1 \cup B_2 \to A^c$, and utilizing the definitions of the central charges in eq. (120), we obtain the following modified formula for computing the holographic entanglement negativity for the bipartite mixed state configuration of a single interval in a thermal $GCFT_{1+1}$ with both central charges non-vanishing:

$$
\mathcal{E} = \lim_{B \to A^c} \frac{3}{16G}\left(2\mathcal{L}_A + \mathcal{L}_{B_1} + \mathcal{L}_{B_2} - \mathcal{L}_{A\cup B_1} - \mathcal{L}_{A\cup B_2}\right),
\tag{139}
$$

where we have defined

$$
\mathcal{L}_X = \mathrm{L}^{extr}_X + \frac{1}{\mu}\Delta\eta_X,
\tag{140}
$$

where $X$ is the specific subsystem under consideration. We now compute the holographic entanglement negativity for a single interval located at the asymptotic null infinity of the geometry described by TMG in FSC. The holographic computations are identical to those in subsection (5.1.3) for the extremal geodesic lengths $\mathrm{L}^{extr}_{ij}$ and the remaining contribution to the holographic entanglement negativity comes from the Chern-Simons term as

$$
\mathcal{E}_{CS} = \frac{c_L}{4}\left(\log\left[\frac{\beta}{\pi\epsilon}\sinh\left(\frac{\pi\phi_{12}}{\beta}\right)\right] - \frac{\pi\phi_{12}}{\beta}\right).
\tag{141}
$$

Together with the Einstein gravity result eq. (58), the total holographic negativity becomes

$$
\mathcal{E} = \frac{c_L}{4}\left[\log\left[\frac{\beta}{\pi\epsilon}\sinh\left(\frac{\pi\phi_{12}}{\beta}\right)\right] - \frac{\pi\phi_{12}}{\beta}\right] + \frac{c_M}{4}\left[\frac{\pi u_{12}}{\beta}\coth\left(\frac{\pi\phi_{12}}{\beta}\right) - \frac{\pi u_{12}}{\beta}\right].
\tag{142}
$$

The above expression for the holographic entanglement negativity exactly matches with the universal part of the complete field theory result obtained in [55] using the replica technique. We may also rewrite eq. (142) in the instructive form eq. (59).

## 7.4 Holographic entanglement negativity for adjacent intervals

Next we turn our attention to the bipartite mixed state configuration of two adjacent intervals in a $GCFT_{1+1}$ with unequal non-vanishing central charges. The holographic entanglement

negativity for the case of Einstein gravity in the bulk was discussed in subsection 5.2. In this subsection we utilize the modified dictionary in eq. (123) to advance a holographic proposal for computing the entanglement negativity for the mixed state configuration of two adjacent intervals living at the null infinity of the geometries described by flat space TMG. From equations (139) and (140), it is easy to see that the expression for the holographic negativity in such scenarios is simply obtained by replacing $L_X^{\text{extr}}$ by $\mathcal{L}_X$, for each subsystem $X$. Therefore our proposal for the entanglement negativity for two disjoint intervals reads (cf. eq. (67)):

$$\mathcal{E} = \frac{3}{16G}\left(\mathcal{L}_{A_1} + \mathcal{L}_{A_1} - \mathcal{L}_{A_1 \cup A_2}\right), \tag{143}$$

with $\mathcal{L}_X$ given in eq. (140).

### 7.4.1 Adjacent intervals at zero temperature

We start with the bipartite mixed state configuration of two adjacent intervals in the vacuum state of the boundary $\text{GCFT}_{1+1}$. To compute the holographic entanglement negativity, we use eq. (128) in our modified holographic entanglement negativity formula eq. (143) to obtain to topological contribution to the entanglement negativity as

$$\begin{aligned} \mathcal{E}_{\text{CS}} &= \frac{3}{16G}\frac{1}{\mu}\left(\Delta\eta_{A_1} + \Delta\eta_{A_2} - \Delta\eta_{A_1 \cup A_2}\right) \\ &= \frac{c_L}{8}\log\left[\frac{t_{12}t_{23}}{\epsilon(t_{12}+t_{23})}\right], \end{aligned} \tag{144}$$

where $\epsilon$ is identified as the UV cut-off. The expression for the total holographic entanglement negativity, after including the Einstein gravity result eq. (69), becomes

$$\mathcal{E} = \frac{c_L}{8}\log\left[\frac{t_{12}t_{23}}{\epsilon(t_{12}+t_{23})}\right] + \frac{c_M}{8}\left(\frac{x_{12}}{t_{12}} + \frac{x_{23}}{t_{23}} - \frac{x_{13}}{t_{13}}\right), \tag{145}$$

which matches exactly with the universal part of the field theory result in [55].

### 7.4.2 Adjacent intervals at a finite temperature

We next compute the holographic entanglement negativity for the bipartite mixed state configuration of two adjacent intervals in a finite temperature $\text{GCFT}_{1+1}$. Here, the boundary theory is defined on an infinite cylinder compactified in the timelike direction leading to a finite temperature $\text{GCFT}_{1+1}$ and the dual gravitational theory is decribed by TMG in FSC geometry. The Chern-Simons contribution to the holographic entanglement negativity, using eq. (143) and eq. (131), is given by

$$\mathcal{E}_{\text{CS}} = \frac{c_L}{8}\log\left[\frac{\beta}{\pi\epsilon}\frac{\sinh\left(\frac{\pi\phi_{12}}{\beta}\right)\sinh\left(\frac{\pi\phi_{23}}{\beta}\right)}{\sinh\left(\frac{\pi(\phi_{12}+\phi_{23})}{\beta}\right)}\right]. \tag{146}$$

The total holographic entanglement negativity after including the Einstein gravity result eq. (70) becomes

$$\begin{aligned} \mathcal{E} = \frac{c_L}{8}\log&\left[\frac{\beta}{\pi\epsilon}\frac{\sinh\left(\frac{\pi\phi_{12}}{\beta}\right)\sinh\left(\frac{\pi\phi_{23}}{\beta}\right)}{\sinh\left(\frac{\pi(\phi_{12}+\phi_{23})}{\beta}\right)}\right] + \frac{c_M}{8}\left[\frac{\pi u_{12}}{\beta}\coth\left(\frac{\pi\phi_{12}}{\beta}\right)\right. \\ &\left.+ \frac{\pi u_{23}}{\beta}\coth\left(\frac{\pi\phi_{23}}{\beta}\right) - \frac{\pi u_{13}}{\beta}\coth\left(\frac{\pi\phi_{13}}{\beta}\right)\right]. \end{aligned} \tag{147}$$

Eq. (147) correctly reproduces the universal part of the result obtained in [55] using field theoretic methods.

### 7.4.3 Adjacent intervals in a finite-sized system

Finally, we focus on the computation of the holographic entanglement negativity for a bipartite mixed state configuration of two adjacent intervals in a $GCFT_{1+1}$ describing a system with finite size. The boundary theory is described by a $GCFT_{1+1}$ on an infinite cylinder compactified in the spatial direction with circumference $L_\phi$. The Chern-Simons contribution to the holographic entanglement negativity is obtained using eq. (132) and eq. (143) as

$$\mathcal{E}_{CS} = \frac{c_L}{8} \log \left[ \frac{\beta}{\pi \epsilon} \frac{\sin\left(\frac{\pi \phi_{12}}{L_\phi}\right) \sin\left(\frac{\pi \phi_{23}}{L_\phi}\right)}{\sin\left(\frac{\pi(\phi_{12}+\phi_{23})}{L_\phi}\right)} \right]. \tag{148}$$

The expression for the total holographic entanglement negativity after including the Einstein gravity result eq. (70) becomes

$$\mathcal{E} = \frac{c_L}{8} \log \left[ \frac{L_\phi}{\pi \epsilon} \frac{\sin\left(\frac{\pi \phi_{12}}{L_\phi}\right) \sin\left(\frac{\pi \phi_{23}}{L_\phi}\right)}{\sin\left(\frac{\pi(\phi_{12}+\phi_{23})}{L_\phi}\right)} \right] + \frac{c_M}{8} \left[ \frac{\pi u_{12}}{L_\phi} \cot\left(\frac{\pi \phi_{12}}{L_\phi}\right) \right.$$
$$\left. + \frac{\pi u_{23}}{L_\phi} \cot\left(\frac{\pi \phi_{23}}{L_\phi}\right) - \frac{\pi u_{13}}{L_\phi} \cot\left(\frac{\pi \phi_{13}}{L_\phi}\right) \right]. \tag{149}$$

This matches exactly with the universal part of the complete field theory result obtained in [55] using replica technique.

## 7.5 Two disjoint intervals in proximity

Finally in this subsection, we compute the holographic entanglement negativity for the bipartite mixed state configuration of two disjoint intervals in a $GCFT_{1+1}$ with both central charges non-vanishing. The entanglement negativity for such configurations involves a four-point function of twist operators with non-zero spin. Therefore to obtain the entanglement negativity via field theoretic methods, we need a semi-classical monodromy analysis of the four-point twist correlator when both the central charges $c_L$ and $c_M$ are non-zero. Note from eq. (121) that $c_L \propto c_M$ when the coupling of the gravitational Chern-Simons term in the bulk dual theory remains finite. Therefore, the previous analysis in subsection 6.1 for a large $c_M$ remains valid and we may obtain a closed form expression of the complete conformal block in the large central charge limit [14]. In the following, we obtain the large central charge expression for the entanglement negativity for two disjoint intervals in a $GCFT_{1+1}$ with both central charges non-vanishing. Subsequently we propose a bulk construction of the holographic entanglement negativity in the dual asymptotically flat geometries incorporating the anomalous effects of the topologically massive gravity.

### 7.5.1 Large central charge negativity and the holographic proposal

In this subsection, we obtain the complete expression for the t-channel Galilean conformal block in the large central large limit for the case where both the central charges of the $GCFT_{1+1}$ are non-zero. Using eqs. (96) and (102), we arrive at the following expression

$$\mathcal{F}_\alpha = \left( \frac{1+\sqrt{T}}{1-\sqrt{T}} \right)^{-\Delta_\alpha} \exp\left[ \chi_\alpha \left( \frac{X}{\sqrt{T}(T-1)} \right) \right]. \tag{150}$$

---

[14]Note that, even if both the central charges $c_L$ and $c_M$ are large, the ratio $\frac{c_M}{c_L} = \mu$ remains finite and therefore the dual anomalous gravitational theory is well defined. Interestingly, in the case of Einstein gravity in the bulk as considered in subsection 6.1, the corresponding limit $\mu \to \infty$ of the TMG action in eq. (119) is reminiscent of the central charge $c_L$ being zero as seen from eq. (121).

To obtain the dominant contribution to the four-point function in eq. (78), we note that the twist operator $\Phi_{n_e}^2$ remains heavy in the replica limit, $\chi_{n_e/2} \to -\frac{c_M}{16}$, $\Delta_{n_e/2} \to -\frac{c_L}{16}$ (cf. footnote 4). Therefore, the partial wave expansion for the four-point twist correlator is dominated by the exchange of $\Phi_{n_e}^2$:

$$
\mathcal{F}_{\Delta_{n_e}^{(2)}, \chi_{n_e}^{(2)}} = \left( \frac{1 + \sqrt{T}}{\sqrt{1 - T}} \right)^{c_L/4} \exp\left( -\frac{c_M}{8} \frac{X}{\sqrt{T}(T-1)} \right). \tag{151}
$$

Finally, using equations (18), (19) and (78), we obtain the entanglement negativity for two disjoint intervals in proximity ($T \to 1$) in the large central charge limit to be

$$
\mathcal{E} \simeq \log\left( \mathcal{F}_{\Delta_{n_e}^{(2)}, \chi_{n_e}^{(2)}} \right) \approx \frac{c_L}{8} \log\left( \frac{1}{1 - T} \right) + \frac{c_M}{8} \frac{X}{1 - T}. \tag{152}
$$

The $t$-channel cross ratios appearing in the above equation may be expressed in terms of the coordinates $(x_i, t_i)$ of the endpoints of the two disjoint intervals in question as

$$
1 - T = \frac{t_{14} t_{23}}{t_{13} t_{24}}, \quad \frac{X}{1 - T} = \frac{x_{13}}{t_{13}} + \frac{x_{24}}{t_{24}} - \frac{x_{14}}{t_{14}} - \frac{x_{23}}{t_{23}}. \tag{153}
$$

Therefore the complete expression for the entanglement negativity for two disjoint intervals $A_1 = [(x_1, t_1), (x_2, t_2)]$ and $A_2 = [(x_3, t_3), (x_4, t_4)]$ in proximity is given by

$$
\mathcal{E} = \frac{c_L}{8} \log\left( \frac{t_{13} t_{24}}{t_{14} t_{23}} \right) + \frac{c_M}{8} \left( \frac{x_{13}}{t_{13}} + \frac{x_{24}}{t_{24}} - \frac{x_{14}}{t_{14}} - \frac{x_{23}}{t_{23}} \right). \tag{154}
$$

We may obtain the large central charge behaviours of the entanglement negativity for a $\mathrm{GCFT}_{1+1}$ describing a finite-sized system as well as a thermal $\mathrm{GCFT}_{1+1}$ by performing suitable conformal maps from the $\mathrm{GCFT}_{1+1}$ plane to the spatially and temporally compactified cylinders respectively, as described before in subsection 6.1.3.

Having described the large central charge behaviour of the entanglement negativity for two disjoint intervals, we now proceed to give a holographic description of such configurations. Utilizing the expression for the two point twist correlator from eq. (7) and eq. (152), the four-point twist correlator appearing in the definition of the entanglement negativity for the two disjoint intervals, have the following large central charge behaviour

$$
\left\langle \Phi_{n_e}(x_1, t_1) \Phi_{-n_e}(x_2, t_2) \Phi_{-n_e}(x_3, t_3) \Phi_{n_e}(x_4, t_4) \right\rangle = \exp\left[ \frac{c_M}{16} \left( \mathcal{L}_{13} + \mathcal{L}_{24} - \mathcal{L}_{14} - \mathcal{L}_{23} \right) \right], \tag{155}
$$

where $\mathcal{L}$ is defined in eq. (114), and we have utilized the relation $\frac{c_M}{c_L} = \mu$. Therefore, for two disjoint intervals $A_1$ and $A_2$ living at the null infinity of the geometries described by TMG in asymptotically flat spacetimes, we propose the following holographic construction for the entanglement negativity

$$
\mathcal{E} = \frac{3}{16G} \left( \mathcal{L}_{A_1 \cup A_s} + \mathcal{L}_{A_s \cup A_2} - \mathcal{L}_{A_1 \cup A_2 \cup A_s} - \mathcal{L}_{A_s} \right), \tag{156}
$$

which tantamounts to replacing $\mathrm{L}^{\text{extr}}$ by $\mathcal{L}$ for the Einstein gravity counterpart in eq. (114), where as before $A_s$ describes another subsystem sandwiched between the two disjoint subsystems in question. In the following, we will apply the above prescription to different bipartite mixed state configurations involving two disjoint intervals in a $\mathrm{GCFT}_{1+1}$ and find agreement with the large central charge results obtained above. Interestingly, all these results may be checked against the İnönü-Wigner contractions of the corresponding $\mathrm{CFT}_2$ results in [29]. This serves as another consistency check of our holographic proposal.

### 7.5.2 Two disjoint intervals at zero temperature

We start with bipartite mixed state configuration of two disjoint intervals in proximity $A_1 = (x_1, t_1)$ and $A_2 = (x_2, t_2)$ in the ground state of a holographic GCFT$_{1+1}$ which is dual to TMG in pure Minkowski spacetime. Using eq. (128) and eq. (23), the holographic entanglement negativity proposal in eq. (156) reproduces the large central charge result in eq. (154).

For comparison, we reproduce the corresponding expression in the context of AdS$_3$/CFT$_2$ from [29]:

$$\mathcal{E} = \frac{c}{8} \log\left(\frac{z_{13} z_{24}}{z_{14} z_{23}}\right) + \frac{\bar{c}}{8} \log\left(\frac{\bar{z}_{13} \bar{z}_{24}}{\bar{z}_{14} \bar{z}_{23}}\right), \tag{157}$$

where we have allowed for unequal central charges $c$ and $\bar{c}$ for the left and right moving sectors, respectively. Now following eq. (1) we take the İnönü-Wigner contractions [40–42]

$$z = t + \epsilon x, \quad \bar{z} = t - \epsilon x, \tag{158}$$

to obtain, up to first order in $\epsilon$,

$$\mathcal{E} = \frac{(c + \bar{c})}{8} \log\left(\frac{t_{13} t_{24}}{t_{14} t_{23}}\right) + \frac{\epsilon(c - \bar{c})}{8} \left(\frac{x_{13}}{t_{13}} + \frac{x_{24}}{t_{24}} - \frac{x_{14}}{t_{14}} - \frac{x_{23}}{t_{23}}\right). \tag{159}$$

Using eq. (11), we see that the GCFT$_{1+1}$ result in eq. (154) is exactly reproduced. This serves as a strong consistency check of our proposal.

### 7.5.3 Two disjoint intervals at a finite temperature

Next, we consider the bipartite mixed state configuration of two disjoint interval in a finite temperature GCFT$_{1+1}$. The boundary theory is living on a cylinder compactified in the timelike direction with circumference $\beta$ and the dual bulk theory is described by global FSC geometry. Using eq. (39) and eq. (132) in the holographic entanglement negativity formula in eq. (156) gives

$$\mathcal{E} = \frac{c_L}{8} \log\left[\frac{\sinh\left(\frac{\pi \phi_{13}}{\beta}\right) \sinh\left(\frac{\pi \phi_{24}}{\beta}\right)}{\sinh\left(\frac{\pi \phi_{14}}{\beta}\right) \sinh\left(\frac{\pi \phi_{23}}{\beta}\right)}\right] + \frac{c_M}{8} \left[\frac{\pi u_{13}}{\beta} \coth\left(\frac{\pi \phi_{13}}{\beta}\right)\right.$$
$$\left. + \frac{\pi u_{24}}{\beta} \coth\left(\frac{\pi \phi_{24}}{\beta}\right) - \frac{\pi u_{14}}{\beta} \coth\left(\frac{\pi \phi_{14}}{\beta}\right) - \frac{\pi u_{23}}{\beta} \coth\left(\frac{\pi \phi_{23}}{\beta}\right)\right]. \tag{160}$$

It is easily verified that the above expression matches perfectly with the large central charge results obtained in subsection 7.5.1.

### 7.5.4 Two disjoint intervals in a finite-sized systems

Finally, we consider the bipartite mixed state configuration described by two disjoint intervals in the proximity in a finite-sized system. Using eq. (132) and eq. (36) we get the holographic entanglement negativity from eq. (156) as

$$\mathcal{E} = \frac{c_L}{8} \log\left[\frac{\sin\left(\frac{\pi \phi_{13}}{L_\phi}\right) \sin\left(\frac{\pi \phi_{24}}{L_\phi}\right)}{\sin\left(\frac{\pi \phi_{14}}{L_\phi}\right) \sin\left(\frac{\pi \phi_{23}}{L_\phi}\right)}\right] + \frac{c_M}{8} \left[\frac{\pi u_{13}}{L_\phi} \cot\left(\frac{\pi \phi_{13}}{L_\phi}\right)\right.$$
$$\left. + \frac{\pi u_{24}}{L_\phi} \cot\left(\frac{\pi \phi_{24}}{L_\phi}\right) - \frac{\pi u_{14}}{L_\phi} \cot\left(\frac{\pi \phi_{14}}{L_\phi}\right) - \frac{\pi u_{23}}{L_\phi} \cot\left(\frac{\pi \phi_{23}}{L_\phi}\right)\right]. \tag{161}$$

This may also be seen to match with the large central charge results in 7.5.1.

# 8 Summary and conclusions

To summarize, we have established a holographic construction to obtain the entanglement negativity for bipartite states in $GCFT_{1+1}$s dual to bulk $(2+1)$-dimensional asymptotically flat Einstein gravity and topologically massive gravity (TMG). For the former the bulk asymptotic symmetry analysis leads to dual $GCFT_{1+1}$s with central charges $c_L = 0, c_M \neq 0$. In this context, we have obtained the holographic entanglement negativity for various bipartite pure and mixed states in a $GCFT_{1+1}$ utilizing our construction. These include the pure state of a single interval dual to a bulk $(2+1)$-dimensional Minkowski spacetime and that in a finite-sized system dual to a bulk global Minkowski orbifold. The corresponding mixed state of a single interval at a finite temperature is dual to a bulk non rotating flat space cosmology described by a null orbifold. Subsequently, the holographic entanglement negativity for the mixed state configuration of two adjacent intervals in a $GCFT_{1+1}$ was computed utilizing our construction. Our results for these bipartite states exactly reproduce the corresponding replica technique results in the large central charge limit.

Following the above computations, we used the geometric monodromy method [67] in the $BMS_3$ field theory to find the large central charge behaviour of the entanglement negativity for the mixed state configuration of two disjoint intervals in the $GCFT_{1+1}$. Utilizing the $\mathcal{M}$ and $\mathcal{L}$ monodromy for each of the two distinct components of the energy-momentum tensor leads to second and third-order differential equations for the four-point twist correlator. Solving these equations, it was possible to obtain the dominant conformal block for the four-point twist correlator in the t-channel describing the intervals in proximity with each other. This leads us to the entanglement negativity for the mixed state configuration under consideration for zero and finite temperature and also finite-sized system described by a $GCFT_{1+1}$ at its large central charge limit. Subsequently we advance a construction to compute the holographic entanglement negativity for this mixed state configuration in zero and finite temperature and also finite-sized system described by a $GCFT_{1+1}$ dual to appropriate bulk gravitational configurations. Interestingly our results exactly match with the corresponding replica technique results in the large central charge limit obtained through the geometric monodromy analysis described above. This constitutes a strong consistency check of our holographic construction for the mixed state configuration in question and may also be extended to the other configurations discussed here in a straightforward fashion. Furthermore we demonstrate that in the limit of the two disjoint intervals being adjacent we retrieve the corresponding holographic entanglement negativity for two adjacent intervals which further demonstrates the consistency of our holographic construction.

Subsequently we have extended our construction to obtain the holographic entanglement negativity for the bipartite states described earlier, in a $GCFT_{1+1}$ with non zero $c_L$ dual to a bulk flat space topologically massive gravity. This describes massive particles with spin propagating in the bulk and also renders both the scaling dimensions for the twist fields to be non zero. Our results for the adjacent and the single intervals match exactly with the corresponding replica technique results in the dual $GCFT_{1+1}$ with both the central charges being non zero. For the mixed state configuration of two disjoint intervals we have extended the monodromy analysis discussed above to the case with a non-zero $c_L$ and subsequently proposed a holographic construction to compute the entanglement negativity for such configurations. Interestingly the results for the holographic entanglement negativity obtained through our construction is identical to the İnönü-Wigner limit for the corresponding replica techniques results for a relativistic $CFT_{1+1}$ which constitutes a consistency check.

It is well known that flat space chiral gravity is a limit of the flat space topologically massive gravity for which the Newton constant $G$ is taken to be infinity and such that the product of $G$ with the coupling constant $\mu$ of the topological term in the action is held fixed. The

corresponding dual GCFT$_{1+1}$ in this case has the other central charge $c_L \neq 0$ and the GCA is identical to the chiral part of a (relativistic) Virasoro algebra. In appendix A utilizing our proposal, we have computed the holographic entanglement negativity for the bipartite pure and mixed state configurations described by single, adjacent, and disjoint intervals in the dual GCFT$_{1+1}$ mentioned above and the results are similar to those obtained earlier for a generic TMG.

In appendix appendix B we have provided various details of the monodromy analysis performed in subsection 6.1.1. The leading order geometric monodromy method utilized to compute the four-point twist correlator associated with the entanglement negativity for two disjoint intervals relies on the exchanged operator being light in the large central charge limit. For generic conformal dimensions of the exchanged operator, the monodromy analysis requires further investigation as the truncation of various quantities up to linear order in the exchanged dimension remains questionable. We have extended the analysis to higher orders in the parameter $\epsilon_\alpha$ and obtained the next to leading order monodromy condition for the three point function. Interestingly, a similar analysis of the four-point twist correlator exactly reproduces the conformal block obtained through the linear analysis. Consequently, we anticipate that the approximate linear solution has the same monodromy properties as the full solution. An İnönü-Wigner contraction of the corresponding relativistic twist correlator in appendix C also hints towards the same. We emphasize that the exact large central charge behavior of the conformal block for the four-point function in question requires a more careful, perhaps non-perturbative, analysis which we leave as a future work.

We would like to emphasize here that our construction described in this work addresses the significant issue of the characterization of mixed state entanglement for a class of dual GCFT$_{1+1}$ in flat space holography. Furthermore it has been shown in the literature that the GCFT$_{1+1}$ dual to a bulk flat space chiral gravity is related to a conformal quantum mechanics (CFT$_1$). This is an extremely interesting open avenue to explore in the future as described by the progress in the corresponding AdS$_2$/CFT$_1$ correspondence. We hope to return to these exciting issues in the near future.

# 9 Acknowledgements

We would like to thank Vinay Malvimat for useful discussions and suggestions which helped in improving this manuscript.

# A Holographic entanglement negativity in flat chiral gravity

In this appendix, we will discuss a special case of flat-space TMG, namely the conformal Chern-Simons gravity (also called flat space chiral gravity) [74]. The dual boundary theory is described by GCA$_2$ with central charges $c_L = 24k$, $c_M = 0$, where $k$ is the Chern-Simons level. The action for conformal Chern-Simons gravity is given by

$$\mathcal{S}_{CSG} = \frac{k}{4\pi} \int d^3x \sqrt{-g} \left[ \varepsilon^{\alpha\beta\gamma} \Gamma^\rho_{\alpha\sigma} \left( \partial_\beta \Gamma^\sigma_{\gamma\rho} + \frac{2}{3} \Gamma^\sigma_{\beta\eta} \Gamma^\eta_{\gamma\rho} \right) \right], \tag{162}$$

with $G \to \infty$, keeping $\mu G = \frac{1}{8k}$ fixed (cf. eq. (119)).

Note that in this case the two-point correlator in (123) only gets a contribution from the Chern-Simon term. In this context, we are looking at a massless spinning particle in the bulk. All the previous analysis in flat-space TMG will now follow with $c_M = 0$ and holographic

entanglement negativity formula for a single interval $A$ becomes

$$\mathcal{E} = \lim_{B \to A^c} \frac{3k\mu}{2} \left( 2\mathcal{X}_A + \mathcal{X}_{B_1} + \mathcal{X}_{B_2} - \mathcal{X}_{A \cup B_1} - \mathcal{X}_{A \cup B_2} \right), \tag{163}$$

where we have defined

$$\mathcal{X} = \frac{1}{\mu} \Delta \eta. \tag{164}$$

Therefore, using eqs. (128) and (163), the holographic entanglement negativity for a single interval in the ground state of a chiral GCFT$_{1+1}$ is obtained as

$$\mathcal{E} = \frac{3}{8G} \mathcal{X}_A = \frac{c_L}{4} \log\left( \frac{t_{12}}{\epsilon} \right). \tag{165}$$

Similarly, we may compute the holographic entanglement negativity for a single interval $A = [(x_1, t_1), (x_2, t_2)]$ at a finite temperature or for finite-sized systems using eq. (131) and eq. (132). The results match exactly with those in the flat-space TMG case as well as the field theory results in [55] with $c_M = 0$ which strongly substantiates our holographic proposals.

Next, we modify our holographic entanglement negativity proposal for two adjacent intervals $A_1 = [(x_1, t_1), (x_2, t_2)]$ and $A = [(x_2, t_2), (x_3, t_3)]$ at the boundary of a manifold accommodating flat chiral gravity:

$$\mathcal{E} = \frac{3k\mu}{2} \left( \mathcal{X}_{A_1} + \mathcal{X}_{A_1} - \mathcal{X}_{A_1 \cup A_2} \right). \tag{166}$$

Using eq. (128) for the spinning contribution in pure Minkowski spacetime, eq. (166) yields the following expression for the holograpic entanglement negativity for two disjoint intervals in the chiral GCFT$_{1+1}$ vacuum:

$$\begin{aligned}
\mathcal{E}_{\text{CS}} &= \frac{3k}{2} \left( \Delta \eta_{A_1} + \Delta \eta_{A_2} - \Delta \eta_{A_1 \cup A_2} \right) \\
&= \frac{c_L}{8} \log\left[ \frac{t_{12} t_{23}}{\epsilon (t_{12} + t_{23})} \right],
\end{aligned} \tag{167}$$

which matches exactly with the $c_M = 0$ version of the dual field theory result in [55]. Similarly, we can obtain the holographic entanglement negativity for adjacent intervals at a finite temperature and for finite-sized systems in the present scenario using eq. (131) and eq. (132).

Finally, for two disjoint intervals $A = [(x_1, t_1), (x_2, t_2)]$ and $B = [(x_3, t_3), (x_4, t_4)]$ in proximity in the chiral GCFT$_{1+1}$ with $c_M = 0$, we write

$$\mathcal{E} = \frac{3k}{2} \left( \mathcal{X}_{13} + \mathcal{X}_{24} - \mathcal{X}_{14} - \mathcal{X}_{23} \right), \tag{168}$$

with $\mathcal{X}$ given in eq. (164). Using eq. (128) and eq. (168), we obtain the holographic entanglement negativity in the ground state to be

$$\mathcal{E} = \frac{c_L}{8} \log\left[ \frac{t_{13} t_{24}}{t_{14} t_{23}} \right]. \tag{169}$$

Similarly, we can obtain negativity for two disjoint intervals at a finite temperature and for finite-sized systems using eq. (131) and eq. (132). Once again the results match with the flat-space TMG results with $c_M = 0$ as well as the corresponding İnönü-Wigner limits of the relativistic field theory results [29].

# B   Next to leading order monodromy

In this appendix, we will perform the geometric monodromy analysis in the next to leading order in the re-scaled conformal dimension $\epsilon_\alpha = \frac{6}{c_M}\chi_\alpha$ of the exchange operator. In particular we will focus only on the monodromy problem associated with the expectation value of the component $\mathcal{M}$ of the energy-momentum tensor in the following. To this end we begin with the differential equation eq. (86) and expand its solutions up to second order in $\epsilon_\alpha$ as follows

$$h_i = h_i^{(0)} + \epsilon_\alpha h_i^{(1)} + \epsilon_\alpha^2 h_i^{(2)}, \quad \mathcal{M} = \mathcal{M}^{(0)} + \epsilon_\alpha \mathcal{M}^{(1)}. \tag{170}$$

To proceed, we recall that the solutions to the first order differential equation in eq. (89) was solved utilizing the method of variation of parameters as

$$h_i^{(1)}(u) = f_{i,1}(u)h_1^{(0)}(u) + f_{i,2}(u)h_2^{(0)}(u), \tag{171}$$

where the functions $f_{i,j}(u)$ may be obtained through the Wronskian of the differential equation as described in [67]. Subsequently, encircling a path enclosing the light operator at $t = T$ the solutions of eq. (86) transform as

$$h_1^{(1)}(u) \to h_1^{(0)}(u) + \left( \oint f_{1,2}'(u) \right) h_2^{(0)}(u),$$

$$h_2^{(1)}(u) \to \left( \oint f_{2,1}'(u) \right) h_1^{(0)}(u) + h_2^{(0)}(u). \tag{172}$$

Specializing to the three point function[15] $\left\langle \Phi_{n_e}(x_1, t_1) \Phi_{n_e}(x_4, t_4) V_\alpha(X, T) \right\rangle$, we note that the expectation value of the energy-momentum tensor is given by

$$\frac{6}{c_M}\mathcal{M}^{(1)}(T; t) = \frac{T}{t(t - T)^2}\epsilon_\alpha. \tag{173}$$

From eqs. (172) and (173) we obtain the monodromy matrix up to first order as

$$M^{(1)} = \begin{pmatrix} 1 & 2iT\pi\epsilon_\alpha \\ \frac{2i\pi\epsilon_\alpha}{T} & 1 \end{pmatrix}, \tag{174}$$

which leads to the monodromy condition in eq. (91).

In the next to leading order, the differential equation reads

$$h_i^{(2)\prime\prime}(t) = -\frac{6}{c_M}\mathcal{M}^{(1)}(T; t)h_i^{(1)}(t). \tag{175}$$

Once again, we solve eq. (175) by variation of parameters as

$$\begin{aligned} h_i^{(2)}(u) &= J_{i,1}(u)h_1^{(1)}(u) + J_{i,2}(u)h_2^{(1)}(u) \\ &= (J_{i,1}f_{1,1} + J_{i,2}f_{2,1})h_1^{(0)}(u) + (J_{i,1}f_{1,2} + J_{i,2}f_{2,2})h_2^{(0)}(u), \end{aligned} \tag{176}$$

where

$$J_{i,1}'(u) = \frac{W_{i,1}}{W}, \qquad J_{i,2}'(u) = \frac{W_{i,2}}{W}, \tag{177}$$

and the Wronskians are given as

$$W_{i,1}(u) = \begin{vmatrix} 0 & \frac{-6}{c_M}M^{(1)}h_i^{(1)} \\ h_2^{(1)} & h_2^{\prime(1)} \end{vmatrix}, \quad W(u) = \begin{vmatrix} h_1^{(1)} & h_1^{\prime(1)} \\ h_2^{(1)} & h_2^{\prime(1)} \end{vmatrix}, \quad W_{i,2}(u) = \begin{vmatrix} h_1^{(1)} & h_1^{\prime(1)} \\ 0 & \frac{-6}{c_M}M^{(1)}h_i^{(1)} \end{vmatrix}.$$

---

[15]Note that the same three-point function appears in the partial wave expansion in eq. (78) and hence provides us with the necessary monodromy condition.

Upon going around the light operator at $t = T$ in a circle the solutions to eq. (175) transform as

$$h_1^{(2)}(u) \to \left[ f_{1,1}(u) + \left( \oint J'_{1,2} \right) f_{2,1}(u) \right] h_1^{(0)}(u) + \left[ f_{1,2}(u) + \left( \oint J'_{1,2} \right) f_{2,2}(u) \right] h_2^{(0)}(u),$$

$$h_2^{(2)}(u) \to \left[ \left( \oint J'_{2,1} \right) f_{1,1}(u) + f_{2,1}(u) \right] h_1^{(0)}(u) + \left[ \left( \oint J'_{2,1} \right) f_{1,2}(u) + f_{2,2}(u) \right] h_2^{(0)}(u). \quad (178)$$

Computing the contour integrals and substituting, we may read off the monodromy matrix in the next to leading order as

$$M^{(2)} = \begin{pmatrix} 4\pi^2 \epsilon_\alpha^2 & 0 \\ 0 & -4\pi^2 \epsilon_\alpha^2 \end{pmatrix}, \quad (179)$$

and the monodromy condition becomes

$$\frac{I_1 - I_2}{2} = -16\pi^4 \epsilon_\alpha^4, \quad (180)$$

where $I_1 = \operatorname{tr} M$ and $I_2 = \operatorname{tr} M^2$ are invariant under global Galilean conformal transformations.

Next we solve the differential equation eq. (175) for the four-point function eq. (78) utilizing the method outlined above. Note that for the four-point function the expectation value of the energy-momentum tensor is given in eq. (83). After solving eq. (175) we compute the monodromy of the solutions by going around the light operators at $t = 1, T$ as described in [67] and retraced above. This leads to the following monodromy matrix

$$M^{(2)} = \begin{pmatrix} 4\pi^2 (T-1)^2 T c_2^2 & 0 \\ 0 & -4\pi^2 (T-1)^2 T c_2^2 \end{pmatrix}. \quad (181)$$

Utilizing the monodromy condition eq. (180) in the next to leading order, we may obtain the auxiliary parameter $c_2$ as

$$c_2 = \epsilon_\alpha \frac{1}{\sqrt{T}(T-1)}. \quad (182)$$

Hence the conformal block for the four-point function in eq. (78) may be obtained as:

$$\begin{aligned} \mathcal{F}_\alpha &= \exp\left[ \frac{c_M}{6} \int c_2 \, dX \right] \\ &= \exp\left[ \chi_\alpha \left( \frac{X}{\sqrt{T}(T-1)} \right) \right] \tilde{\mathcal{F}}(T). \end{aligned} \quad (183)$$

Remarkably, this is exactly the same conformal block eq. (93) obtained in subsection 6.1.1 confining ourselves to first order in the parameter $\epsilon_\alpha$. The above analysis may be extended to higher orders in $\epsilon_\alpha$ in a similar manner.

The analysis in this appendix hints towards the fact that the monodromy method works at each order in the expansion parameter. We may interpret this as follows: the approximate solution to the differential equation eq. (86) worked out in subsection 6.1.1 is able to pick up the same monodromy while circling around the light operators, as would the complete solution.

# C İnönü-Wigner contraction

In this appendix we will further analyze the large central charge behavior of the four-point twist correlator in eq. (73) relevant to the entanglement negativity for two disjoint intervals in

proximity in a GCFT$_{1+1}$. In particular, we will show that the proximity limit of the dominant conformal block in eq. (151) may be obtained by performing the İnönü-Wigner contractions [40–42] of the corresponding relativistic result obtained in the context of $AdS_3/CFT_2$ in [23]. Note that the parametric contractions given in eq. (158) may alternatively be written in terms of the coordinates describing the relativistic $CFT_{1+1}$ as

$$z \to t + \epsilon x, \qquad \bar{z} \to t - \epsilon x. \tag{184}$$

The central charges of the $GCA_2$ are related to those of the parent relativistic theory as

$$c_L = c + \bar{c}, \quad c_M = \epsilon(c - \bar{c}). \tag{185}$$

Next, we recall that the four-point twist correlator associated with the mixed state configuration of two disjoint intervals in proximity in a relativistic $CFT_{1+1}$ is given by the following expression in the large central charge limit [23, 25, 29]:

$$\lim_{n_e \to 1} \left\langle \mathcal{T}_{n_e}(z_1) \bar{\mathcal{T}}_{n_e}(z_2) \bar{\mathcal{T}}_{n_e}(z_3) \mathcal{T}_{n_e}(z_4) \right\rangle = (1-x)^{-c/4}, \tag{186}$$

where $x = \frac{z_{12} z_{34}}{z_{13} z_{24}}$ is the $CFT_{1+1}$ cross ratio. If we allow for unequal central charges for the left and right moving sectors the above expression has the natural generalization

$$\lim_{n_e \to 1} \left\langle \mathcal{T}_{n_e}(z_1) \bar{\mathcal{T}}_{n_e}(z_2) \bar{\mathcal{T}}_{n_e}(z_3) \mathcal{T}_{n_e}(z_4) \right\rangle = (1-x)^{-c/8}(1-\bar{x})^{-\bar{c}/8}. \tag{187}$$

Utilizing eq. (184), we now write the $CFT_{1+1}$ cross ratios in terms of those in the $GCFT_{1+1}$ as

$$x \to T\left(1 + \epsilon \frac{X}{T}\right), \qquad \bar{x} \to T\left(1 - \epsilon \frac{X}{T}\right). \tag{188}$$

Now performing the İnönü-Wigner contraction of the cross ratios as given in eq. (188), we obtain for the corresponding correlator in the GCFT$_{1+1}$ as

$$\lim_{n_e \to 1} \left\langle \Phi_{n_e}(x_1, t_1) \Phi_{-n_e}(x_2, t_2) \Phi_{-n_e}(x_3, t_3) \Phi_{n_e}(x_4, t_4) \right\rangle \to \left[1 - T\left(1 + \epsilon \frac{X}{T}\right)\right]^{-c/8} \left[1 - T\left(1 - \epsilon \frac{X}{T}\right)\right]^{-\bar{c}/8}. \tag{189}$$

where the cross-ratios $X$ and $X/T$ are defined as

$$T = \frac{t_{12} t_{34}}{t_{13} t_{24}}, \qquad \frac{X}{T} = \frac{x_{12}}{t_{12}} + \frac{x_{34}}{t_{34}} - \frac{x_{13}}{t_{13}} - \frac{x_{24}}{t_{24}}. \tag{190}$$

Expanding upto linear order in $\epsilon$ and using eq. (185), the above expression reduces to

$$\lim_{n_e \to 1} \left\langle \Phi_{n_e}(x_1, t_1) \Phi_{-n_e}(x_2, t_2) \Phi_{-n_e}(x_3, t_3) \Phi_{n_e}(x_4, t_4) \right\rangle \approx \left(\frac{1}{1-T}\right)^{c_L/4} \exp\left(-\frac{c_M}{8} \frac{X}{T-1}\right). \tag{191}$$

Remarkably, this expression matches exactly with that obtained through the geometric monodromy method in eq. (152). This provides further support towards the validity of the monodromy analysis up to leading order in the exchanged dimension.

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
