# Peer review of "Entanglement Negativity in Flat Holography"

_SciPost Physics, doi:SciPost Phys. 12, 074 (2022)_

## Round 1 · Referee Report · Anonymous (Referee 1) · 2021-8-29

Report

This paper studied extensively the holography negativity in flat holography, ranging form single interval to multiple intervals, from Einstein gravity to TMG. The paper is rather long and contains lots of small technical details. Without going through all the details, I find that the strategies and results look reasonable to me. Therefore, I would recommend the publication of the paper.
  • validity: -
  • significance: -
  • originality: -
  • clarity: -
  • formatting: -
  • grammar: -

Author:  Gautam Sengupta  on 2021-10-09  [id 1830]

(in reply to Report 1 on 2021-08-29)

We would like to thank the referee for the comment and for recommending publication.

---

## Round 1 · Referee Report · Anonymous (Referee 2) · 2021-9-7

Strengths

The authors consider various cases and perform several checks of their results.

Weaknesses

Unclear what new physics the results obtained teach us.

Report

A potentially interesting result of this article is the large central charge negativity of two disjoint intervals. To compute it, the authors use the standard CFT monodromy method applied to the case of Galilean conformal symmetry with the help of [67].

They provide formula (104) for the Galilean conformal block associated with certain four point functions (relevant in the computation eg of entanglement, negativity etc). The result is valid in the large central charge limit, and when the dominant contribution comes from a light operator (and the monodromy is computed around a light operator).

In section 6.1.3, the authors apply (104) for computing the entanglement negativity of disjoint intervals. However, they simultaneously claim that the dominant contribution in this case comes from an operator that remains heavy in the large rental charge limit - like in [25]. I would be grateful if the authors could further clarify this point, before considering the article for publication.

Requested changes

See above.

  • validity: -
  • significance: good
  • originality: low
  • clarity: -
  • formatting: -
  • grammar: -

Author:  Gautam Sengupta  on 2021-10-09  [id 1831]

(in reply to Report 2 on 2021-09-07)

We would like to thank the referee for the comments and raising the issue described in the report. Our detailed response to the comment and the clarification of the issue is appended. We have included this clarification in our revised manuscript as suggested by the referee.

In this article we consider a four point twist correlator in the GCFT$_{1+1}$ and wish to compute the monodromy in the $t$-channel by going around a loop enclosing the light operators situated at $(0,1)$ and $(X,T)$ which fuse together. By performing the $\mathcal{M}$- and $\mathcal{L}$-monodromies, we computed the generic conformal block for an exchange operator of conformal dimension $\chi_\alpha$ in eq. (104). The conformal block for the four-point function of the twist operators is obtained perturbatively in the parameter $\epsilon_{\alpha}$ which is related to the conformal dimension of the corresponding exchange operator which is hence required to be light. Nevertheless, as described in reference [13, 25] of our revised manuscript, in the large central charge limit the dominant contribution to equation (78) comes from the exchange operator with the lowest conformal dimension which can be obtained from the leading order term in the operator product expansion (OPE) in the $s$- and $t$-channels. For our case the dominant contribution to the entanglement negativity for the two disjoint intervals in proximity (for the $t$-channel) may be obtained from that of the conformal block corresponding to the exchange of the lightest operator.

Specifically, we consider the OPE of light operators located at the positions $[(x_2,t_2)\,,\,(x_3,t_3)]$ for the $t$-channel as shown in equation (105) in our updated submission. The fusion channels include the light operators $\Phi_{-n_e}(x_2,t_2)$, $\Phi_{-n_e}(x_3,t_3)$ and $\Phi_{n_e}(x_1,t_1)$, $\Phi_{n_e}(x_4,t_4)$ for the $t$-channel. It is evident from equation (105) that the leading order contribution arises from the exchange operator $\Phi^2_{\pm n_e}$ for the $t$-channel as given in equation (107). In this case the operator $\Phi^2_{\pm n_e}$ is the lightest of the exchange operators with the lowest conformal dimension for the $t$-channel. This serves as a justification for the monodromy analysis as described in ref [23, 25] in the framework of the usual relativistic CFT, and the use of equation (104) although $\Phi^2_{\pm n_e}$ actually remains heavy in the replica limit. We have added a discussion of this subtle issue and provided the relevant OPE expansions in equation (105) in the beginning of subsection 6.1.3 of our revised submission.

We would also like to point out that the holographic entanglement negativity for the disjoint intervals (in proximity) has also been obtained from the extremal entanglement wedge cross section in arXiv 2106.14896 recently. The corresponding entanglement negativity matches with our result computed via monodromy analysis which further serves as a consistency check for our computations.

---

## Round 2 · Referee Report · Anonymous (Referee 2) · 2021-10-27

Report

I thank the authors for their response. I would however appreciate some further clarification by the authors. To make my question more precise:

To determine the conformal block eq(93) of the article (and hence eq(105)), one needs to impose certain monodromy conditions. In the relativistic case, there is only one condition, namely that the trace of the monodromy matrix should be related in a specific way to the conformal dimension of the operator exchanged in the respective channel.

Analogous conditions in the Galilean case for the HHLL (heavy-heavy-light-light) four point function and for exchanged/intermediate operators of small conformal dimension, were found in ref[67]. Namely that: trM=2, accompanied by eq(97) of this article.

Can perhaps the authors comment as to why these conditions are valid here when the external operators are all light (in the replica limit) and the operator exchanged is not light?
If I understand well the authors claim that the exchanged operator is the lightest of all exchanged operators. I agree with this, but it was my impression that the derivation was based on the assumption of an exchanged operator with small conformal dimension with respect to the central charge in the large central charge limit.
  • validity: -
  • significance: -
  • originality: -
  • clarity: -
  • formatting: -
  • grammar: -

Author:  Gautam Sengupta  on 2021-11-23  [id 1967]

(in reply to Report 1 on 2021-10-27)

We would like to thank the referee for the comment. The clarification sought by the referee is described below and included in our revised submission and detailed in two additional appendices B and C.

In this article we consider a four point twist correlator in the GCFT$_{1+1}$ where in the replica limit, all of the external operators are light but the exchange operator in the conformal block expansion remains heavy in the large central charge limit. We agree with the referee that the perturbative analysis in the expansion parameter $\epsilon_\alpha=\frac{6}{c_M}\chi_\alpha$ requires further investigation when the exchange operator is heavy. We would like to emphasize that even if $\epsilon_\alpha$ is not infinitesimally small, we may still perform a series solution to the Fuchsian differential equation given in eq. (86) of our revised manuscript. For the case when $\chi_\alpha$ is associated with a heavy exchange operator, we need to be more careful and in principle should keep track of the full series solution.

The first subtlety faced in such a series solution is that whether the monodromy condition given in eq. (91) of our revised manuscript has the correct form. We emphasize that eq. (91) reflects the monodromy condition only in the first order in the expansion parameter $\epsilon_\alpha$. We have added the derivation of the first order monodromy condition in appendix B of our revised manuscript. In particular, the monodromy matrix in the leading order is given in eq. (174), from which eq. (91) follows.

Since the truncation of the series solution up to the first order remains questionable, we have performed the next to leading order monodromy analysis also in appendix B of our revised manuscript and the corresponding monodromy condition is given in eq. (180). Interestingly, for the four point function in question, the next to leading order monodromy analysis shows that the auxiliary parameter $c_2$ and hence the dominant conformal block $\mathcal{F}_\alpha$ has exactly the same form as obtained through the first order analysis. This is reported in equations (182, 183) and discussed briefly in footnote 10 on page 23 of our revised manuscript. The analysis in appendix B hints towards the fact that the monodromy method works at each order in the expansion parameter. We may interpret this as follows: the approximate solution to the differential equation is able to pick up the same monodromy while circling around the light operators, as would the complete solution. This provides a strong substantiation of our result for the dominant conformal block. These subtleties have been briefly discussed in footnote 10 on page 23 of our revised manuscript.

Furthermore, in appendix C of our revised submission, we have shown that the four-point twist correlator in a (1+1)-dimensional GCFT may be obtained by performing the Inonu-Wigner contractions of the corresponding relativistic result in the context of $AdS_3/CFT_2$ obtained in ref [23,25] of our revised manuscript. Remarkably, the result matches exactly with that obtained through the geometric monodromy method. This provides further support towards the validity of the monodromy analysis up to leading order in the exchanged dimension. Furthermore as pointed out in our earlier response an independent holographic crosscheck from the bulk entanglement wedge cross section as described in arXiv: 2106.14896 also confirms our results for the mixed state configuration in question.

We have also discussed the above issues in the summary section of our revised manuscript. In addition, we would like to point out a small typographical error in eq. (150) of our revised submission regarding a sign discrepancy that has now been corrected.

---

## Round 2 · Author Response

We have examined the reports of the referees for our submission scipost\_202107\_00037v1 entitled ``Entanglement Negativity in Flat Holography". We would like to thank the referees for the comments and the issue raised . The first referee has recommended publication and has suggested no changes. Our detailed response to the second referee's comment and the clarification of the issue raised by the second referee has been described in the reply to the second referee 's report. We have included this clarification in our revised manuscript as suggested by the second referee.

---

## Round 2 · List of Changes

The detailed response to the referee' s comments and issues raised has been described in the author' s reply section.

---

## Round 3 · Referee Report · Anonymous (Referee 1) · 2021-12-6

Report

I recommend the publication of this paper on SciPost Physics. The paper overall is OK, although the originality and significance are not so impressive.

---

## Round 3 · Referee Report · Anonymous (Referee 2) · 2021-12-22

Report

I am happy with the authors' responses to my question and added sections in the article.

---

## Round 3 · Author Response

We have examined the appended report of the second referee for our submission scipost\_202107\_00037v2 entitled " Entanglement Negativity in Flat Holography" seeking further clarification of the issue. Our detailed response to the referee's comment and further clarification of the issue is provided in the comment section in reply to the referee. We have included the clarifications in our revised manuscript as suggested by the referee and detailed the issue further in two new appendices B and C.

---

## Round 3 · List of Changes

We have provided the complete list of changes in the authors response section.

---

## Editorial Decision

published